evolution/environmental science/ecology

Atlantic salmon, heart morphology, liver, aquaculture, allometry, sexual dimorphism

**Author for correspondence:**
William Bernard Perry
e-mail: w.perry@bangor.ac.uk

# Disentangling the effects of sex, life history and genetic background in Atlantic salmon: growth, heart and liver under common garden conditions

William Bernard Perry[1], Monica F. Solberg[2], Christopher Brodie[3], Angela C. Medina[4], Kirthana G. Pillay[1], Anna Egerton[1], Alison Harvey[2], Simon Creer[1], Martin Llewellyn[5], Martin Taylor[6], Gary Carvalho[1] and Kevin A. Glover[2,7]

[1]Molecular Ecology and Fisheries Genetics Laboratory, School of Biological Sciences, Bangor University, Bangor, Gwynedd LL57 2UW, UK
[2]Population Genetics Research Group, Institute of Marine Research, PO Box 1870, Nordnes 5817, Bergen, Norway
[3]Mariani Molecular Ecology Laboratory, School of Natural Sciences and Psychology, Liverpool John Moores University, Liverpool L3 5UX, UK
[4]School of Microbiology, Food Science and Technology Building University College Cork, Cork T12 TP07, Ireland
[5]Institute of Biodiversity, Animal Health and Comparative Medicine, University of Glasgow, Glasgow G12 8QQ, UK
[6]School of Biological Sciences, University of East Anglia, Norwich, NR4 7TJ, UK
[7]Institute of Biology, University of Bergen, Bergen, Norway

  WBP, 0000-0001-9596-3333; AH, 0000-0001-8422-8763; MT, 0000-0002-3858-0712; GC, 0000-0002-9509-7284

Livestock domestication has long been a part of agriculture, estimated to have first occurred approximately 10 000 years ago. Despite the plethora of traits studied, there is little understanding of the possible impacts domestication has had on internal organs, which are key determinants of survival. Moreover, the genetic basis of observed associated changes in artificial environments is still puzzling. Here we examine impacts of captivity on two organs in Atlantic salmon (*Salar salar*) that have been domesticated for approximately 50 years: heart and liver, in addition to growth. We studied multiple families of wild, domesticated, $F_1$ and $F_2$ hybrid, and backcrossed strains of *S. salar* in replicated common garden tanks during the freshwater and marine stages of

development. Heart and liver weight were investigated, along with heart morphology metrics examined in just the wild, domesticated and $F_1$ hybrid strains (heart height and width). Growth was positively linked with the proportion of the domesticated strain, and recombination in $F_2$ hybrids (and the potential disruption of co-adapted gene complexes) did not influence growth. Despite the influence of domestication on growth, we found no evidence for domestication-driven divergence in heart or liver morphology. However, sexual dimorphism was detected in heart morphology, and after controlling for body size, females exhibited significantly larger heart weight and heart width when compared with males. Wild females also had an increased heart height when compared with wild males, and this was not observed in any other strain. Females sampled in saltwater showed significantly larger heart height with rounder hearts, than saltwater males. Collectively, these results demonstrate an additive basis of growth and, despite a strong influence of domestication on growth, no clear evidence of changes in heart or liver morphology associated with domestication was identified.

# 1. Introduction

The process of livestock domestication has long been a part of agriculture and is estimated to have first occurred approximately 10 000 years ago with the domestication of goats (*Capra hircus*) in the western highlands of Iran [1]. Domestication in fish, however, was first observed in species of tilapia (*Oreochromis niloticus*) (4000 years ago) and carp (*Cyprinus carpio*) (8000 years ago) [2–4], with the aquaculture-driven domestication of salmonids not starting until the 1970s [5,6]. Studies examining the impacts of domestication have typically focused on features such as bone shape [7], behaviour [8] and genetics [9]. Assessment of internal organs in domesticated taxa, including structural heart morphology, has received little attention (but see [10,11]). A higher proportion of studies investigating the impact of domestication on heart morphology have done so in fish [12–14]. The relatively recent wide-scale domestication of fish [15,16], and in particular salmonids, offers a valuable insight into the early stages of domestication on organs such as the heart and liver. Moreover, domestication has proven an insightful model for exploring fundamental tenets of the extended evolutionary synthesis [17,18].

Atlantic salmon (*Salmo salar*), an economically major aquaculture species, exhibits an anadromous life history, transitioning between freshwater and marine habitats. Throughout this diverse life history, there is an underlying mosaic of evolutionary selection pressures, acting on all aspects of biology, from morphology to behaviour. While all aspects are important to individual fitness, certain traits can affect survival to a far greater magnitude. Understandably, vital organs provide a key role in survival and success of fish, including the brain, liver and heart, and this is particularly true of vagile taxa, where energy demands and complex migratory patterns underpin distribution and abundance. Movement in challenging habitats and long-distance migrations characterize the life history of Atlantic salmon where individuals can migrate thousands of kilometres across their lifetime [19]. Not only are great distances covered, but individuals tackle strong freshwater currents en route to natal breeding grounds, along with numerous topographic barriers, such as waterfalls and rapids. A component required for such intense and protracted activity, in combination with musculature and body-shape, is strong cardiac function [20].

Although heart morphology has been shaped by natural selection, in recent decades, the evolutionary trajectory of wild fish has changed due to domestication in aquaculture [21]. Atlantic salmon aquaculture was initiated in the early 1970s, and cultured fish that have been artificially selected for economically important traits for approximately 13 generations now form the basis of the industry [22]. Captive propagation under unnatural conditions has resulted in a wide variety of genetic differences between domesticated and wild salmon (reviewed by Glover et al. [21]), the most notable of which, growth, now displays a sevenfold increase in domesticated salmon [23–31]. In addition to growth, directional selection has also impacted traits such as delayed maturation [25,32,33] and fillet quality [34], while also permitting inadvertent or hitchhiking selection resulting in trait shifts such as predator avoidance behaviour [35], sexual morphology [36] and stress susceptibility [37].

In addition to the traits discussed above, it has also been suggested that domestication may drive genetic changes in Atlantic salmon heart morphology and liver weight [12,13,38]. These studies have revealed differences in heart morphology between farmed and wild fish, with farmed fish displaying more rounded hearts, a hallmark of a more sedentary fish species; along with deposition of fat around

the heart. Likewise, farmed fish such as cod (*Gadus morhua*) have been shown to have heavier livers than their wild counterparts, with the development of fatty deposits [13,39]. However, as the fish were not reared under common environmental conditions, it remains challenging to disentangle the relative impacts of genetics and environment on observed differences. Other traits, such as spot patterns, also show large differences between wild and farmed Atlantic salmon, and common garden studies have demonstrated that this is primarily a plastic response [40]. Here, we investigated whether genetic differences in heart morphology and liver weight could be detected between domesticated and wild salmon when reared in a common garden design. We also investigated growth differences among strains, with second-generation crosses, providing a contrast with the influence of domestication on heart morphology.

Each year, thousands or hundreds of thousands of domesticated salmon escape into the wild, and where it has been studied, extensive introgression of domesticated salmon has been observed in many wild populations [41]. Although improvements in infrastructure have reduced the incidence of reported escapees, monitoring programmes demonstrate that there remain large numbers of domesticated escapees on spawning grounds of some rivers [42]. Such wild–farmed interactions are a cause for concern given the shifts in traits of domesticated individuals and wild–domesticated hybrids, as domesticated and hybrid fish have reduced fitness in the wild [43–46], thereby compromising the genetic integrity, and long-term fitness of wild populations [21]. Understanding changes in the biology of domesticated fish is therefore not only relevant to understanding the processes and changes during domestication, they are also fundamental to our understanding of how wild–farmed interactions impact on wild populations and communities [47,48].

# 2. Material and methods

## 2.1. Overall experimental design

In order to investigate potential genetic differences in growth between domesticated, wild, $F_1$ hybrid, $F_2$ hybrid and backcrossed salmon, seven synchronously produced experimental strains, each with multiple pedigree-controlled families, were reared in a common garden environment from hatching onwards and later identified using a genetic parentage analysis. In addition to parentage, genetic sex was also identified using microsatellite markers. A common garden environment ensures that all experimental strains are reared under the same environmental conditions by rearing them together. The same fish were used to assess heart morphology (including adjusted heart (AH) height, AH width and heart width–height residuals (WHR)) in the domesticated, $F_1$ hybrid and wild strains (table 1). Fish were sampled both as aged 1+ smolts in freshwater after being reared in replicated tanks, and as post-smolts at aged 2+ in replicated saltwater tanks.

## 2.2. Experimental fish

The fish used here were produced at the Matre Research Station (60°52′26.4″ N, 5°35′09.0″ E), with fertilization of gametes occurring on 1 December 2015 following a wild, domesticated, hybrid, backcross common garden design, implemented over a decade by the Institute of Marine Research [25,28,29,49,50]. Seven experimental strains, represented by 36 different families and produced by 18 male and 18 female parent broodstock, were used in this experiment, including: a wild strain (from the river Figgjo 58.819° N, 5.559° E), a domesticated strain (developed by Mowi and domesticated for approx. 13 generations), hybrid FM (Figgjo (♀) × Mowi (♂)), hybrid MF (Mowi (♀) × Figgjo (♂)), $F_2$ hybrids, wild × hybrid backcrosses and domesticated×hybrid backcrosses (table 1). A subset of wild, domesticated and reciprocal $F_1$ hybrids were then used to investigate heart morphology. Individuals from all strains were mixed into four replicate tanks, with the first two tanks representing the freshwater stage of the life cycle euthanized between the dates 25–28 April 2017. The second two tanks representing the saltwater stage of the life cycle were euthanized a year later between the dates 23–27 April 2018, after having been transferred to saltwater in June 2017. To identify the genetic background of fish in the common garden post-sampling, a genetic parentage analysis was conducted.

During the freshwater life stage, fish were reared in a flow-through system of four replicated octagonal tanks, located in an enclosed outbuilding. Final rearing tanks were 3 m wide and 1.25 m deep with a volume of 6300 l, and a continuous flow rate (60 l min$^{-1}$), supplied with freshwater from several sources surrounding the research station at Matre. Incoming water was passed through 15 m

**Table 1.** Number of fish used for wet weight, heart weight and heart morphology measures such as AHH, AHW and heart WHR. Counts are broken down by life stage (freshwater or saltwater), experimental strains and sex (male = ♂ and female = ♀). The seven experimental strains are included both in terms of their broader types, as well as the specific strains, as well as the number of families that represent those strains, as well as the geographical location of the wild strain.

| origin | | | | wet weight | | heart weight | | heart morphology (AHH, AHW, WHR) | |
| --- | --- | --- | --- | --- | --- | --- | --- | --- | --- |
| type | strain | number of families | location | freshwater | saltwater | freshwater | saltwater | freshwater | saltwater |
| domesticated | Mowi | 6 | | ♂ = 98 ♀ = 80 | ♂ = 72 ♀ = 93 | ♂ = 43 ♀ = 48 | ♂ = 34 ♀ = 42 | ♂ = 8 ♀ = 15 | ♂ = 12 ♀ = 15 |
| hybrid | Figgjo (♀) × Mowi (♂) | 3 | | ♂ = 59 ♀ = 38 | ♂ = 52 ♀ = 32 | ♂ = 34 ♀ = 24 | ♂ = 24 ♀ = 16 | ♂ = 14 ♀ = 1 | ♂ = 13 ♀ = 9 |
| | Mowi (♀) × Figgjo (♂) | 3 | | ♂ = 57 ♀ = 39 | ♂ = 53 ♀ = 43 | ♂ = 23 ♀ = 23 | ♂ = 16 ♀ = 16 | ♂ = 5 ♀ = 3 | ♂ = 6 ♀ = 8 |
| F₂ hybrid | | 6 | | ♂ = 95 ♀ = 76 | ♂ = 92 ♀ = 98 | ♂ = 50 ♀ = 43 | ♂ = 45 ♀ = 43 | | |
| wild | Figgjo | 6 | 58°81′ N, 5°55′ E | ♂ = 103 ♀ = 81 | ♂ = 72 ♀ = 76 | ♂ = 54 ♀ = 52 | ♂ = 29 ♀ = 28 | ♂ = 13 ♀ = 12 | ♂ = 10 ♀ = 13 |
| wild backcross | | 6 | | ♂ = 90 ♀ = 78 | ♂ = 86 ♀ = 87 | ♂ = 36 ♀ = 41 | ♂ = 33 ♀ = 38 | | |
| domesticated backcross | | 6 | | ♂ = 84 ♀ = 78 | ♂ = 92 ♀ = 79 | ♂ = 47 ♀ = 44 | ♂ = 45 ♀ = 30 | | |

high concrete header tanks and filtered through a 40 µm filtration unit before entering individual tanks. Fish from two of the four freshwater replicates then went on to rearing in two flow-through saltwater tanks. Tanks were 5 m wide, 1.1 m deep, had a volume of 15 600 l and a flow rate of 170–200 l min$^{-1}$, supplied with water from the surrounding fjord. Both freshwater and saltwater tanks were lit artificially, starting with a 24 h light regime during first feeding, with the photoperiod simulating that of Bergen post first feeding. The temperature range of the water during the experiment was 3°C–14.2°C. Fish were fed on a diet of pellets produced by Skretting Nutra Olympic (Cheshire, UK).

Fish were euthanized using an anaesthetic overdose of MS-222. All researchers working directly with the experimental animals had undergone Norwegian Food Safety Authority (NFSA) training, in compliance with experimentation involving live animals, included in the Animal Welfare Act (Norway). As the fish in the experiment were not further manipulated, and exposed only to standard rearing conditions, no specific research permit was required.

## 2.3. Parentage analysis and genetic sex

Genomic DNA was extracted from alcohol-preserved fin-clip samples using the Qiagen DNeasy® 96 Blood & Tissue Kit, followed by a multiplex PCR which amplified six microsatellite loci; SsaF43 [GenBank: U37494] [51], Ssa197 [GenBank: U43694.1] [52], SSsp3016 [GenBank: AY372820], MHCI [53] and MHCII [54]. Genetic sex was identified by the presence of the *sdY gene* [55,56]; if the presence of exon 2 and 4 were detected, the individual was designated as male. An ABI Applied Biosystems ABI 3730 Genetic Analyser was used for fragment analysis, the outputs of which were used to call genotypes in GeneMapper (Applied Biosystems, v. 4.0). Further details are outlined by Solberg *et al.* [28,37].

## 2.4. Body weight

Wet body weight, hereon referred to as body weight, was measured on 1272 fish in the freshwater life stage, and 1146 fish in the saltwater life stage (table 1). In addition to body weight, fork length (to the neared 1 mm) was also measured using a standard measuring board. A fin-clip for DNA-family identification was taken from all individuals during sampling.

A linear mixed effect model was used to assess factors influencing weight with the R package 'lme4' [57]. The response variable for the linear mixed effect model was log10 transformed body weight. The full models contained the fixed factors: life stage, sex, strain and all two-way interactions, along with random factors: date of dissection, dam, sire, family nested in strain and tank. Family was nested within strain in order to account for non-independence in the data, as families are unique to their corresponding strain. The full model was simplified using the 'step' function within 'lme4' through automatic backward elimination removing fixed terms and random factors which did not contribute to the model. Analysis of variance type III sum of squares with Satterthwaite approximation for degrees of freedom allowed for the generation of *p*-values using the R package 'lmerTest' [58]. Estimated marginal means and pairwise comparisons between means were calculated using the R package 'emmeans' [59] while using the Tukey's multiple-testing adjustment, generating 95% confidence intervals, and degrees of freedom using the Kenward–Roger approximation.

## 2.5. Dissection and heart measurements

Heart ventricles were freed from the thoracic cavity by cutting along the bulbus arteriosus and pulling away the atrium (figure 1*a*). The ventricle was used to assess overall heart morphology due to it being the largest most muscular chamber, pumping blood entering from the atrium through the bulbus arteriosus (figure 1*c*). The ventricle is also the area of the heart that has been investigated most thoroughly in previous studies [12,60]. Livers were freed from the main body cavity by making incisions along the bile duct and hepatic blood vessels. Hearts and livers were immediately weighed and placed in 4% buffered paraformaldehyde (PFA) for fixation. Heart samples taken from freshwater life stages remained in 4% PFA for 9 days. Due to the larger size of the heart samples taken from the saltwater life stage, these remained in 4% PFA for 42 days. Once fixed, tissue samples were then moved to 70% ethanol for long-term storage.

Height and width of the heart ventricle were measured using callipers, as outlined in figure 1*b*, and were based on previous studies assessing heart morphology in Atlantic salmon [12]. The height of the heart is defined here as the line extending from the base of the bulbus arteriosus to the apex of the heart, and the width is defined as the widest length of the ventricle parallel to the base. Heart

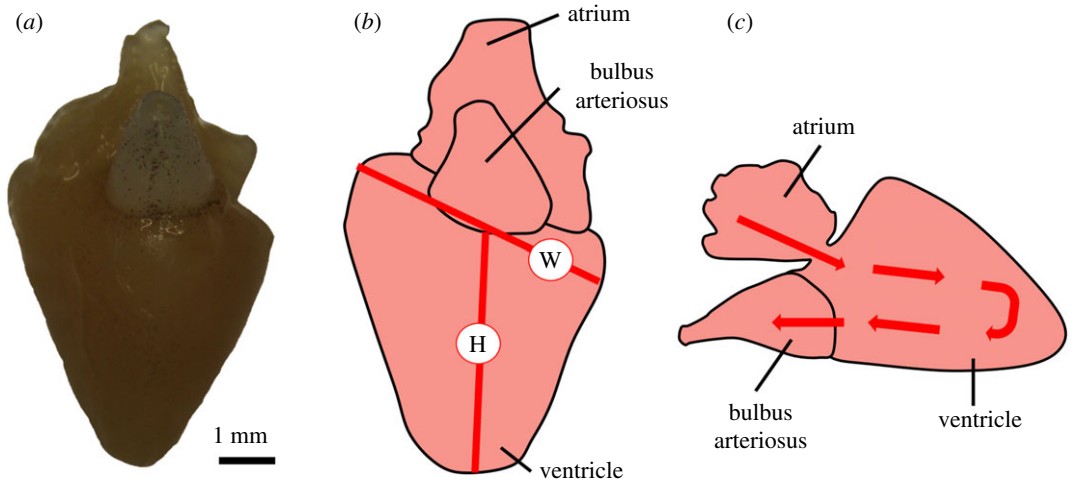

**Figure 1.** Anatomy of the Atlantic salmon (*Salmo salar*) heart, as demonstrated by (*a*) photograph microscopy of a heart from the freshwater life stage, as well as diagrammatically (*b,c*). Red lines in (*b*) labelled with the letters H and W represent the measurements heart height and heart width, respectively. Arrows displayed in (*c*) show the direction of blood flow within the single circulatory system of the teleost heart.

ventricle height and width are herein referred to as heart height and heart width. All heart dissection and measurements were taken by one person without knowledge of the genetic background of the fish.

## 2.6. Secondary morphological measures

A linear regression was constructed between $\log_{10}$ heart height and $\log_{10}$ fork length, as well as $\log_{10}$ heart width and $\log_{10}$ fork length. Residuals from this linear regression were then used as fork length-adjusted heart height (AHH) and fork length-adjusted heart width. A linear regression was also constructed between $\log_{10}$ heart weight and $\log_{10}$ body weight, as well as $\log_{10}$ liver weight and $\log_{10}$ body weight. Residuals from this linear regression were then used as body weight-adjusted heart weight and body weight-adjusted liver weight (ALW). AH weight was used instead of relative heart mass (RHM). Finally, a linear regression was constructed between $\log_{10}$ heart width and $\log_{10}$ heart height, residuals from this linear regression, herein described as WHR. Measurements relating to AHH, AH width and heart WHR were not conducted on backcross or $F_2$ hybrid strains. Backcross and $F_2$ hybrid strains were used for AH weight and ALW, however.

## 2.7. Analysis of heart and liver measurements

When analysing AH weight, AHH, AH width, WHR and ALW each were used as separate response variables in five linear mixed effect models (LME) that were constructed using the R package 'lme4' [57]. The full models contained the fixed factors: life stage, sex, strain and all two-way interactions, along with random factors: date of dissection, dam, sire, family nested in strain and tank. Models were then simplified using the 'step' function in the package 'lme4'. Estimated marginal means and pairwise comparisons between means were calculated using the selected models and the R package 'emmeans' [59], as previously described in the body weight section.

In addition to the analysis of our data, a proof of concept was also included, contrasting the outcome of two different methods in adjusting for allometry: (i) use of residuals from a regression between the measure of interest and a measure of body size, and (ii) divisional indexes whereby the measure of interest is divided by a measure of body size. We use the saltwater subset of our data, adjusting heart weight using the two methods, resulting in the residual-based AH weight, and the division-based RHM. RHM is calculated using the following formula: heart weight (g)/body weight (g) × 100. Both AH weight and RHM were included as response variables in a linear regression with body weight. To further show the problems of using RHM, neutrally simulated data whereby random simulations ($n = 990$) (using the range, standard deviation and mean of the observed heart and body weight data) were also produced. The randomly produced dataset was then used to calculate a neutrally simulated RHM, which was subsequently used as the response variable in a regression with the neutrally

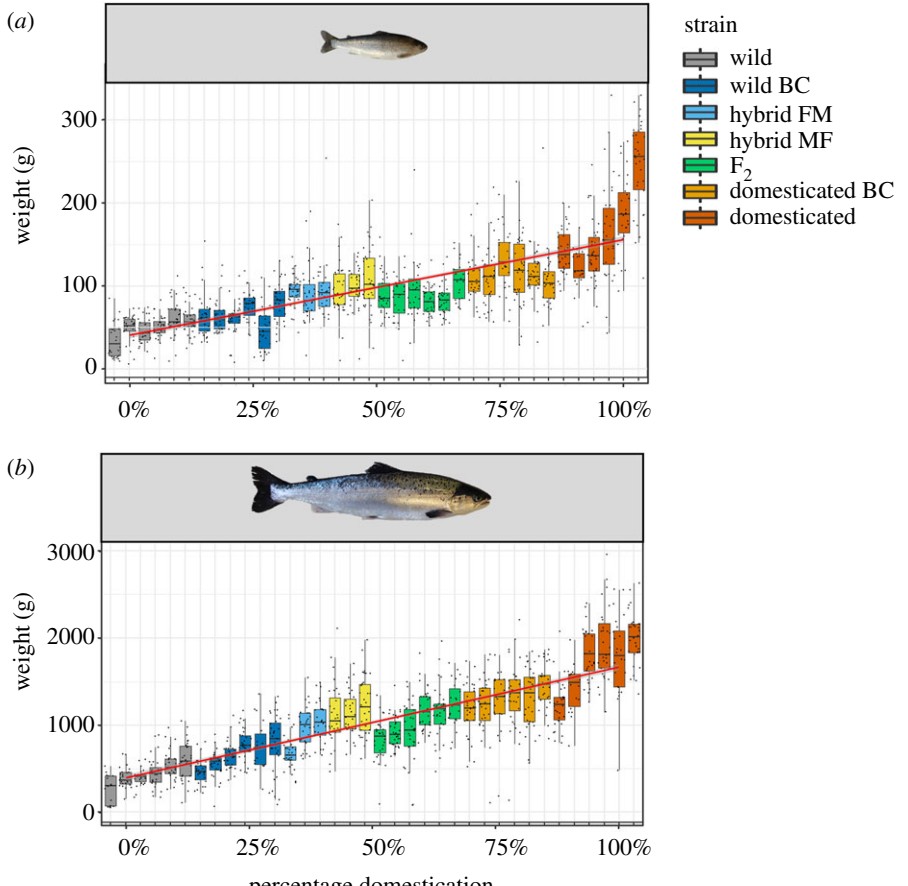

**Figure 2.** Boxplots of wet weight in grams between both (*a*) freshwater and (*b*) saltwater individuals, further broken down into the families that make up the seven experimental strains (wild, wild backcross (wild BC), hybrid FM (Figgjo (♀) × Mowi (♂)), hybrid MF (Mowi (♀) × Figgjo (♂)), F$_2$ hybrids, domesticated backcross (domesticated BC) and domesticated (Mowi)), as shown by the different colours. Additionally to the boxplots of wet weight per family, there is also a linear regression, as shown in red, that was run between the percentage levels of domestication, including 0% (wild), 25% (wild BC), 50% (F$_1$ and F$_2$ hybrids), 75% (domesticated BC) and 100% (domesticated). Percentage level of domestication used here is calibrated to our experimental design, with our domesticated strain being 100% domesticated, and our wild strain being 0% domesticated. Intermediate levels of domestication are produced through different combinations of hybridization, be it direct hybrids (50% domesticated, 50% wild), wild backcrosses (25% domesticated, 75% wild) or domesticated backcrosses (75% domesticated, 25% wild).

simulated body weight. Finally, using the entire dataset, the same full model for AH weight (as described above) was fitted with RHM as the response variable. After the step function, the final model for RHM contained the fixed factors sex and life stage as well as the random factors family nested in strain and dissection date.

# 3. Results

## 3.1. Body weight

Body weight increased in line with the proportion of the domesticated strain within each of the seven experimental groups (LME Strain: $F_{6,28} = 38.70$, Sum Sq $= 5.84$, $p < 0.01$) (figure 2). The final model contained the fixed factors strain, sex, life stage, an interaction term between strain and life stage, an interaction term between sex and strain, as well as the random factors family nested in strain, and tank. Significant pairwise differences ($p \leq 0.05$) in mean body weight were observed in 15/21 pairwise comparisons (electronic supplementary material, table S1). Thus, both F$_1$ and F$_2$ hybrids displayed intermediate body weight to the wild and domesticated strains, while both backcrossed variants displayed body weight intermediate between hybrids and their respective wild or domesticated strain.

There was also a significant effect of sex on body weight (LME Sex: $F_{1,2021} = 11.28$, Sum Sq $= 0.283$, $p < 0.01$), in addition to a significant interaction between sex and strain (LME Sex $*$ Strain: $F_{6,2022} = 5.50$, Sum

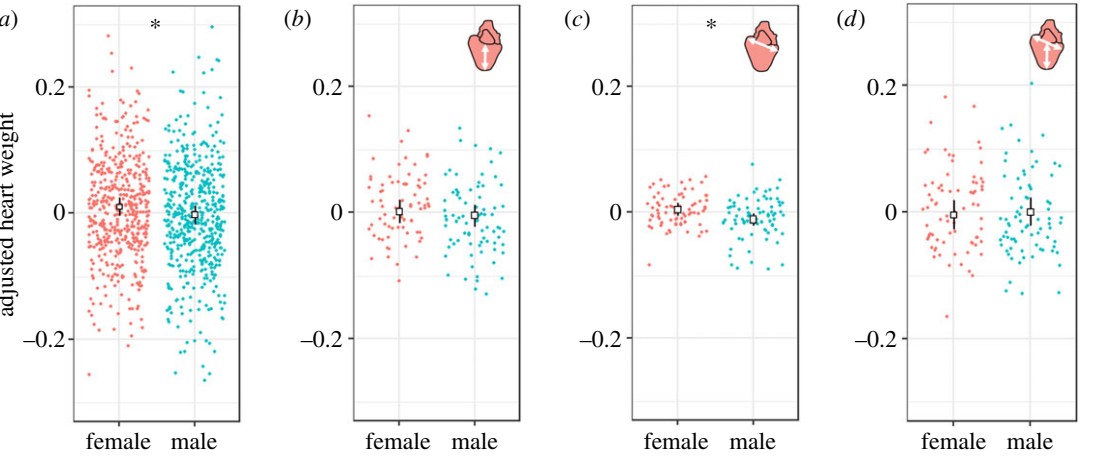

**Figure 3.** Estimated marginal means and confidence intervals from the LME for (*a*) AH weight, (*b*) AHH, (*c*) AH width and (*d*) WHR. Results are split between sexes. Significant differences between the sexes are indicated with an asterisk. It should also be highlighted that there are significant interaction terms with sex for (*b*) AHH and (*d*) WHR.

Sq = 0.83, $p < 0.01$). A sex difference was only seen in the wild strain ($t_{2024} = 4.54$, $p < 0.01$) and the wild backcross strain ($t_{2028} = 4.81$, $p < 0.01$), whereby females had larger body weight than males. All other differences between sexes within a strain were non-significant ($p > 0.05$) (electronic supplementary material, table S2). As would be expected, life stage showed a strong influence over body weight (LME Life stage: $F_{1,2} = 2031.39$, Sum Sq = 51.05, $p < 0.01$). In addition, there was a significant interaction between strain and life stage (LME Strain * Life stage: $F_{6,2018} = 7.28$, Sum Sq = 1.10, $p < 0.01$) (electronic supplementary material, table S3). A full breakdown of estimated marginal means per strain and per life stage with standard deviation and standard error can be found in electronic supplementary material, table S4.

## 3.2. Liver weight

There were no significant effects of life stage, sex, strain (as highlighted in the boxplot of body weight ALW between strains in electronic supplementary material, figure S2) or any of the interaction terms on body weight ALW.

## 3.3. Heart morphology

### 3.3.1. Adjusted heart weight

Strain and life stage showed no significant effect on AH weight. The final model contained sex as a fixed factor, as well as family nested in strain and date of dissection as random factors. Sex was shown to be the only significant factor influencing AH weight (LME Sex: $F_{1,965} = 6.28$, Sum Sq = 0.04, $p = 0.01$), with female fish having significantly larger AH weight (estimated mean = 0.011) to their male counterparts (estimated mean = −0.002) ($t_{964} = 2.50$, $p = 0.01$) (figure 3*a*). Variation in AH weight due to family background (s.d. = 0.03) and dissection date (s.d. = 0.02) were detected and controlled for as random factors in the linear mixed effect model.

### 3.3.2. Adjusted heart height and adjusted heart width

Strain did not show a significant effect on AHH (LME Strain: $F_{3,13} = 1.57$, Sum Sq = 0.010, $p = 0.86$). A LME was run with response variable AHH, and after the step function, the final model contained the fixed factors: life stage, sex, strain, an interaction term between sex and life stage, an interaction term between sex and strain, and the random factor family nested in strain. Life stage showed a significant effect on AHH (LME Life stage: $F_{1,140} = 5.80$, Sum Sq = 0.013, $p = 0.02$) (figure 4), with a significant difference in estimated means between freshwater (estimated mean = −0.011) and saltwater (estimated mean = 0.009) ($t_{140} = 2.39$, $p = 0.02$); with a further significant interaction between life stage and sex (LME Life stage * Sex: $F_{1,137} = 7.96$, Sum Sq = 0.017, $p < 0.01$) (figure 4). Sex alone was not significant (LME Sex: $F_{1,137} = 0.57$, Sum Sq = 0.001, $p = 0.45$) (figure 3*b*). The significant interaction between sex

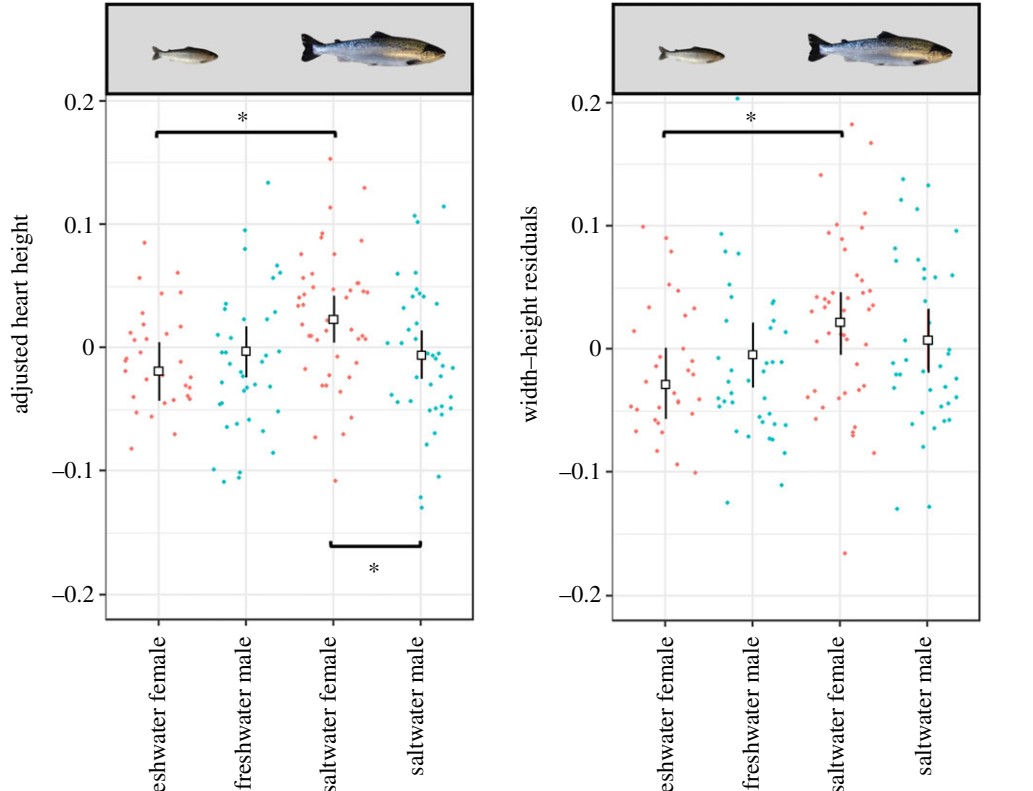

**Figure 4.** Estimated marginal means and confidence intervals from the LME for AHH and WHR. Results are split into freshwater and saltwater life stages, as well as between sexes. Sample numbers for AHH and WHR between sex, life stage and genetic background are outlined in table 1. Significant ($p < 0.05$) pairwise comparisons are shown with brackets and an asterisk.

and life stage was driven by a significant difference between freshwater females and saltwater females ($t_{140} = 3.53$, $p < 0.01$). Saltwater females had a significantly larger AHH (estimated mean = 0.024) than freshwater females (estimated mean = −0.019). The difference between freshwater males and saltwater males was not significant ($t_{137} = 0.27$, $p > 0.05$). There was also a significant difference between saltwater females and saltwater males ($t_{139} = 2.85$, $p = 0.03$), whereby saltwater females had significantly larger AHH (estimated mean = 0.024) than saltwater males (estimated mean = −0.006). Finally, there was a significant interaction term between sex and strain (LME Sex * Strain: $F_{3,135} = 3.72$, Sum Sq = 0.024, $p = 0.01$), driven entirely by the difference between male and female wild fish ($t_{142} = 3.441$, $p = 0.02$), where female wild fish have a larger AHH (estimated mean = 0.010) than male wild fish (estimated mean = −0.041). Variation in AHH due to family background (s.d. = 0.02) was detected and controlled for as a random factor in the linear mixed effect model.

Strain did not show a significant effect on AH width, and neither did life stage. The same full model for AHH was applied to the response variable AH width, and after the step function, the final model contained the fixed factor sex and the random factor family nested in strain. Sex showed a significant effect on AH width (LME Sex: $F_{1,145} = 12.74$, Sum Sq = 0.009, $p < 0.01$), with female fish showing significantly larger AH width (estimated mean = 0.0051) when compared with males (estimated mean = −0.0111) (figure 3$c$). Variation in AH width due to family background (s.d. = 0.01) was detected and controlled for as a random factor in the linear mixed effect model.

### 3.3.3. Heart width–height residuals

Strain did not show a significant effect on heart WHR. The same full model as for AH weight, AHH and AH width was applied to the response variable WHR, and after the step function, the final model contained the fixed factor life stage, sex and an interaction term between sex and life stage, as well as the random factor family nested in strain. A significant effect on WHR was seen for life stage (LME Life stage: $F_{1,142} = 10.51$, Sum Sq = 0.032, $p < 0.01$), with an increase in WHR seen in saltwater individuals when compared with freshwater; this is equivalent to a lower H : W ratio, and thus a

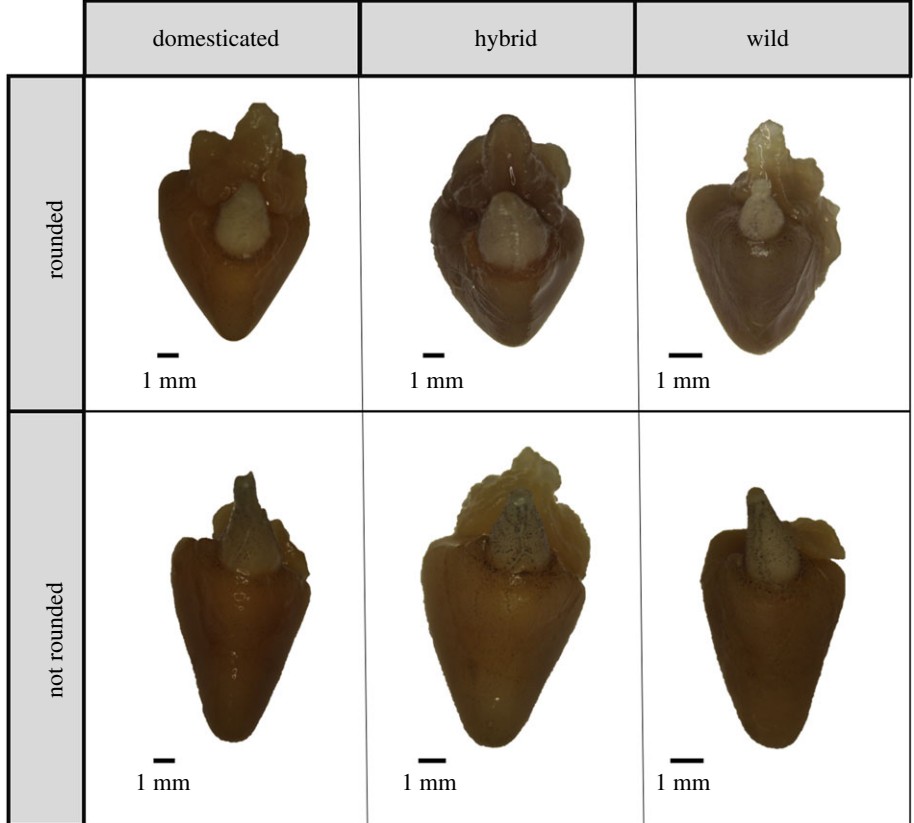

**Figure 5.** Examples of difference in heart shape in domesticated, hybrid and wild fish. As discussed in previous literature, there has been an interest in how round the ventricle is, as defined by the relationship between the height and width. Here, as in Poppe *et al.* [12], rounded hearts are those which have more equal height and width measurements, which is characterized by a lower H : W ratio (closer to 1), or a higher width–height residual (WHR). What is demonstrated here is the rounded and not-rounded morphology can be found in domesticated, hybrid and wild strains.

more rounded heart in saltwater individuals, as is described by Poppe *et al.* [12]. We acknowledge here that although individuals with an increased WHR are described as rounded due to overall heart shape, as in Poppe *et al.* [12], the tip of the ventricle appears less rounded in round hearted individuals (figure 5).

Although sex alone was not significant (LME Sex: $F_{1,143} = 0.27$, Sum Sq = 0.001, $p = 0.61$) (figure 3$d$), there was a significant interaction term between life stage and sex in the final model (LME Life stage $^*$ Sex: $F_{1,139} = 4.03$, Sum Sq = 0.012, $p = 0.05$) (figure 4) which was driven by the significantly larger WHR seen in saltwater females (estimated mean = 0.021) when compared with freshwater females (estimated mean = −0.029) ($t_{143} = 3.60$, $p < 0.01$), while males showed no significant difference between life stages. Variation in WHR due to family background (s.d. = 0.03) was detected and controlled for as a random factor in the linear mixed effect model.

## 3.4. Summary

Strain was only seen as a significant effect in the linear mixed effect model assessing body weight (table 2). In addition to this, family variation was also a significant random effect in all LME, other than the model assessing ALW. Finally, sex was included in all models, apart from the model assessing ALW, and was a significant factor independently, or due to an interaction with another factor, in all instances.

## 3.5. Proof of concept: residuals versus division when adjusting for body size

The regression between AH weight and fish weight for the saltwater individuals used in this study was not significant ($R^2 = 0.006$, $F_{1,425} = 2.57$, $p = 0.11$); the regression between RHM and fish weight, however,

**Table 2.** Linear mixed effect model summary table, assessing the response variables: body weight, body weight ALW, body weight AH weight, fork length AHH, for k length AH width and heart WHR. The effect of fixed factors: life stage, sex, strain, their interaction terms, as well as random factors: date of dissection, dam, sire, family nested in strain and tank are displayed. Included are factors that contributed to the final model ($\checkmark$), and those that were removed by the 'step' function within 'lme4' (X). Significant factors are in bold, with $F$-values and degrees of freedom included for all fixed factors that contributed to the final model, as well as likelihood ratio test (LRT) for random effects.

| | fixed factors | | | | | | | random | | | | |
|---|---|---|---|---|---|---|---|---|---|---|---|---|
| | life stage | sex | strain | life stage*sex | life stage*strain | sex*strain | life stage* strain*sex | date of dissection | dam | sire | strain:family | tank |
| body weight | $\checkmark$ **$F_{1,2}$ = 2031.39**, **$p < 0.01$** | $\checkmark$ **$F_{1,2021}$ = 11.28**, **$p < 0.01$** | $\checkmark$ **$F_{6,28}$ = 38.70**, **$p < 0.01$** | X | $\checkmark$ **$F_{6,2018}$ = 7.28**, **$p < 0.01$** | $\checkmark$ **$F_{6,2022}$ = 5.50**, **$p < 0.01$** | X | X | X | X | $\checkmark$ **LRT = 204.43**, **$p < 0.01$** | $\checkmark$ **LRT = 14.03**, **$p < 0.01$** |
| adjusted liver weight (ALW) | X | X | X | X | X | X | X | X | X | X | X | X |
| adjusted heart weight (AH weight) | X | $\checkmark$ **$F_{1,965}$ = 6.28**, **$p < 0.01$** | X | X | X | X | X | $\checkmark$ **LRT = 16.52**, **$p < 0.01$** | X | X | $\checkmark$ **LRT = 46.71**, **$p < 0.01$** | X |
| adjusted heart height (AHH) | $\checkmark$ **$F_{1,140}$ = 5.80**, **$p = 0.02$** | $\checkmark$ **$F_{1,137}$ = 0.57**, $p = 0.45$ | $\checkmark$ $F_{3,13}$ = 1.57, $p = 0.85$ | $\checkmark$ **$F_{1,137}$ = 7.96**, **$p < 0.01$** | X | $\checkmark$ **$F_{3,135}$ = 3.72**, **$p < 0.01$** | X | X | X | X | $\checkmark$ **LRT = 11.76**, **$p < 0.01$** | X |
| adjusted heart width (AH width) | X | $\checkmark$ **$F_{1,145}$ = 12.74**, **$p < 0.01$** | X | X | X | X | X | X | X | X | $\checkmark$ **LRT = 8.478**, **$p < 0.01$** | X |
| heart width–height residuals (WHR) | $\checkmark$ **$F_{1,142}$ = 10.51**, **$p < 0.01$** | $\checkmark$ **$F_{1,143}$ = 0.27**, $p = 0.61$ | X | $\checkmark$ **$F_{1,139}$ = 4.03**, **$p = 0.05$** | X | X | X | X | X | X | $\checkmark$ **LRT = 18.82**, **$p < 0.01$** | X |

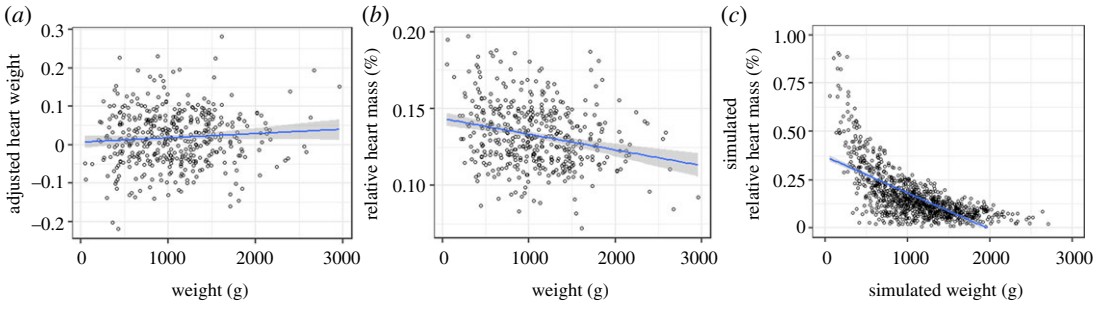

**Figure 6.** Regression plots between saltwater fish weight and two methods of removing the impact of fish weight on heart weight. The first is what is used in this study, (*a*) AH weight, which are residuals from a regression between fish weight and heart weight. The second is what has been widely used in previous studies, (*b*) RHM. Additionally, there is (*c*) neutrally simulated data, whereby random simulations (using the range, standard deviation and mean of the observed heart and body weight), displays the inverse relationship between RHM and fish weight.

was significant ($R^2 = 0.060$, $F_{1,423} = 26.81$, $p < 0.001$) (figure 6*a,b*). The regression between naturally simulated RMV and body weight was also significant ($R^2 = 0.252$, $F_{1,988} = 334.4$, $p < 0.001$) (figure 6*c*). When using the entire dataset, a significant effect on RHM was seen for life stage (LME Life stage: $F_{1,15} = 61.83$, Sum Sq = 0.050, $p < 0.001$), with a significant increase in RHM seen in freshwater individuals when compared with saltwater. Sex was also a significant effect (LME Sex: $F_{1,967} = 10.64$, Sum Sq = 0.009, $p < 0.01$), with a significant increase in RHM seen in males when compared with females.

## 4. Discussion

To our knowledge, this is the first study to investigate differences in heart morphology between domesticated and wild Atlantic salmon reared in common garden conditions, and, the first to present extensive growth data on a full matrix of $F_2$-generation crosses and backcrosses. Based upon the described experimental conditions, we observed large differences in body weight between the seven strains investigated, positively linked to the proportion of the domesticated strain. However, despite domestication playing a major positive role in body weight and thus growth, we found no evidence of domestication-driven divergence in heart morphology. Based on these results, we conclude that while domestication has strongly impacted Atlantic salmon growth capacity, primarily following an additive genetic model [61], no detectable effects in heart morphology have arisen through approximately 13 generations of domestication.

### 4.1. Growth

The impact of domestication on growth and body size using common garden experiments has been examined multiple times in Atlantic salmon, under differing experimental manipulations, which show the effect of strong directional selection placed on growth since the original breeding programmes approximately 13 generations ago [23–31]. Collectively, these studies demonstrate that under identical hatchery conditions with unlimited access to food, domesticated salmon grow faster than their wild counterparts (typically two- to fourfold) due to directional selection for this trait in breeding programmes. One proposed explanation for increased growth rates in domesticated fish is genetically increased appetite [62]. The present study has the novel addition of both backcrossed variants (i.e. backcrossed to domesticated and wild fish), as well as $F_2$ hybrids. Examining backcross and $F_2$ hybrids allows us to better understand the impact of introgression on phenotypes beyond one generation, involving processes such as recombination; as would be experienced by the progeny of escapees in the wild. The observed relationship between mean strain growth and its proportion of domesticated strain, from 0–25–50–75–100%, demonstrate the primarily additive effect of genetic background; i.e. backcrossed strains showed an intermediate body size between the reciprocal $F_1$ hybrids and their respective generator wild or domesticated strains (figure 2). Furthermore, the $F_2$ hybrid strain showed a similar body weight to the $F_1$ hybrid strains (figure 2). Therefore, as the genetic contribution of domesticated and wild backgrounds is still 50% in both the $F_1$ and $F_2$ hybrid strains, only being recombined in $F_2$ hybrids, this shows that this recombination, and potential disruption of co-adapted gene complexes, has not influenced growth rates.

Finally, an interaction between sex and strain was shown to be significant across both life stages. The difference in body weight between males and females was being driven by the sex differences within the wild and wild backcross strains, with larger body weight in females. Once the proportion of domesticated strain increased above 25%, differences in body weight between the sexes disappeared. Sexual dimorphism in size has been observed in Atlantic salmon previously, showing larger body size in immature smolt males when compared with females, the opposite to what is described here; however, this previous experiment was conducted on a domesticated strain, rather than on fish from a wild genetic background [63].

## 4.2. Liver

We saw no significant effects of life stage, sex, strain or any of the interaction terms on body weight ALW. Previous studies have shown that farmed cod (*Gadus morhua*) have significantly heavier livers than wild counterparts, although this is most likely due to environmental plasticity [13] and the development of enlarged fatty livers when cod are fed high-energy lipid-rich diets [39]. Despite no significant differences in ALW between wild and domesticated fish found here, there are studies that control for environmental variation which show that gene expression in the liver of salmonids has been impacted by domestication [64,65]. Therefore, it is possible that there could be fine-scale histological differences caused by domestication that are not detected by looking at weight alone, providing an avenue for further study.

## 4.3. Heart morphology

### 4.3.1. Life stage

There were significant effects of life stage on two, albeit linked, aspects of heart morphology: AHH and WHR, which were also combined with a significant interaction term with sex in both cases (figure 4). Saltwater individuals were seen to have larger AHH which was driven by the larger AHH of saltwater females over freshwater females, and the larger AHH of saltwater females over saltwater males. Saltwater individuals were also seen to have a higher WHR, with larger WHR values associated with a more rounded heart shape, which again was driven by the increase seen in saltwater females over freshwater females, but also the increase seen between saltwater males over freshwater females. One possible explanation, as to why the larger AHH is observed in saltwater females, is that this feature could help with the metabolic load associated with gamete production. However, further study to understand how larger AHH and rounding of the heart relate to stroke volume and overall cardiac performance would be crucial to exploring this hypothesis.

### 4.3.2. Sexual dimorphism

Females were seen to have a significantly larger AH weight (measure used instead of RHM) and AH width when compared with males (figure 3), as well as saltwater females showing larger AHH when compared with saltwater males and freshwater females (figure 4). Sexual dimorphism in heart weight in salmonids has been examined previously, with studies often reporting larger heart sizes in males or no sexual dimorphism at all; however, of these studies, few have done so in a statistically robust manner. For example, studies have either not taken into account variation in body size [66], or body size is considered through simple divisional indexes such as RHM [60,67–69]. The use of ratios and divisional indexes has been widely criticized in the literature, as they are inadequate for removing size correlations from morphological data [70,71]. The divisional index employed in previous studies to adjust heart weight for body weight is referred to as RHM.

We show here, with our proof of concept, that the regression between AH weight and fish weight was not significant; however, the regression between RHM and fish weight, was significant (figure 6a,b), demonstrating that the method does not remove the influence of body size effectively. Randomly simulated data (figure 6c) also demonstrates the inverse relationship that RHM has with fish weight, and that when fitted with a linear regression, produces a significant negative correlation under the null model. Finally, when using RHM as a response variable in the linear mixed effect model, sex was a significant effect, but with a significant increase in RHM seen in males when compared with females, the opposite trend to that seen when using the more statistically rigorous AH weight. The evidence provided here raises concern that previous studies reporting larger male heart sizes are doing so due to

body size being inadequately controlled for [60,67,69]. In addition to this, it adds to previous evidence which demonstrates that division-based indexes are not appropriate for removing the effects of allometry.

If results from previous studies are generically applicable, that is male salmonids have a larger heart weight than females relative to body weight, then results here indirectly demonstrate that sexual dimorphism in these previous studies is due to environmental factors linked with sex, rather than an intrinsic feature. Environmentally driven sexual dimorphism with males having a larger heart weight would suggest that males take part in activities that require higher oxygen demands than females. We would therefore not pick up these differences, as our common experimental design controls for environmental conditions. The intrinsically larger AH weight and AH width seen here in females could be linked with metabolically expensive activities that are not dependant on the environment, such as preparation for oogenesis, if indeed a larger AH weight and AH width does relate to increased functional capacity. To test the effect of environment further, reciprocal 'wild' common garden studies must be conducted, in addition to the hatchery common garden study outlined here. Monitoring the levels of activity in the fish, in addition to experimental groups of differing aerobic training in a laboratory setting, could also elucidate the role of environment on heart sexual dimorphism.

We show here that there was also an interaction term between sex and strain for AHH. Only in the wild strain were there differences in AHH between sexes, with female fish having a significantly larger AHH than males. The sexual dimorphism in AHH disappears in all non-wild strains, and so it can be assumed that the sexual dimorphism has been selected against, either directly or indirectly in aquaculture. A possible scenario whereby this could be envisaged is through wild females undertaking a metabolically expensive process that is no longer required in aquaculture, and so it has been selected against in the trade-off with artificial selection for growth. Such an activity could be linked with sexual selection, which is completely removed in the aquaculture setting, and has been reported to have changed other morphological features in Atlantic salmon [36]. Alternatively, as reproductive success of females in the wild is dependent on (i) oogenesis, (ii) access to territories, and (iii) nest quality [72], and as oogenesis takes place in aquaculture, by deduction, it is possible that a longer AHH could be beneficial in finding access to territories, or for building and maintaining quality nests. To fully understand why there is a sexual dimorphism in wild fish, more wild strains should be examined, as it is possible this dimorphism could differ between populations.

## 4.4. Strain

We found no effect of genetic background on the heart morphology metrics used here, unlike in a related study by Poppe et al. [12]. The lack of differences between domesticated and wild Atlantic salmon seen in this study is not an isolated account, however, with other aspects of cardiac health in Atlantic salmon not differing between the two strains [73]. One difference between this study and the previous study by Poppe et al. [12] is that here we use the width–height residual (WHR) metric instead of the H : W ratio. The problems with divisional indexes and ratios (such as the H : W ratio) are discussed above, in the case of RHM, with literature outlining how they can contribute to spurious self-correlations [74]. We therefore adopted the use of WHR here, to prevent type 1 error.

A second difference between our study and those before, is that here the environment is controlled in a common garden design, whereas Poppe et al. [12] used fish reared in the wild and in an aquaculture setting. Therefore, the suboptimum heart morphology in farmed salmon described by Poppe et al. [12] could be due to environmental plasticity, which bodes well for the cardiac health of farmed fish, and suggests that appropriate cardiac training could prevent suboptimum heart morphology. Finally, different points in the salmon life cycle could play a role. Fish sampled in this previous study were of sizes ranging from 0.5 to 6.4 kg, of which the wild fish could have been multiple sea winter fish and could have also spawned multiple times. Spawning multiple times requires multiple upstream migrations, as well as more competition for mates and nest building, which could affect heart morphology. Likewise, even if the fish had not spawned before, they may have spent a greater length of time at sea or may be at different levels of sexual maturation; all of which could also impact on heart morphology.

## 5. Conclusion

We describe here the largely additive effect of domestication on growth rate, which increases with the percentage of the line that has been domesticated. Additionally, recombination in the $F_2$ hybrids did

not disrupt this additive effect, suggesting that co-adapted gene complexes do not play a vital role in growth. Despite the clear changes in growth caused by domestication, we do not see any clear changes in heart morphology between wild, domesticated and hybrid strains. However, sex and life stage were seen to influence aspects of heart morphology. Sexual dimorphism was seen in AH weight and AH width, with females showing larger hearts relative to body size, with future scope to try and link this with function and metabolically expensive processes such as oogenesis. Similarly, AHH and WHR were also seen to be sexually dimorphic, but this was driven by saltwater females, with again, scope to investigate associations between biomechanics and heart function with processes such as oogenesis. Finally, the observed sexual dimorphism in AHH measurements, with females having larger AHH values, restricted to wild fish only, suggests that domestication may have relaxed selection for sexual dimorphism through direct or indirect artificial selection, as has been seen with other features of Atlantic salmon morphology [36].

Ethics. Those working directly with the experimental animals had also undergone Norwegian Food Safety Authority (NFSA) training, as is required with experimentation involving animals that are included in the Norwegian Animal Welfare Act (2010).

Data accessibility. You can find all morphometric data used in this study in the electronic supplementary material, along with the R code used for the analyses.

Authors' contributions. W.B.P. organized sampling, DNA parentage analysis, sexing, fixation of hearts, measurement of hearts, data analysis and drafting of the original manuscript. K.A.G. and M.F.S. produced the fish for the experiment. M.F.S. was involved in constructing the mixed effect models. C.B., A.C.M. and K.G.P. were involved in sampling, including dissections, weight measurements and tissue samples. A.E. and A.H. were involved in the DNA parentage and sexing of the fish. All authors, including those mentioned and S.C., M.L. and M.T., were involved in the conception and design of the experiment and contributed to data interpretation and editing the final draft of the manuscript. K.A.G. and G.C. managed the project.

Competing interests. The authors declare no competing interests.

Funding. This work was funded by the Norwegian Research Council project INTERACT (grant no. 200510), and the UK Natural Environment Research Council (NERC) Envision doctoral training programme.

Acknowledgements. We would like to thank the staff at the Institute of Marine Research's Matre Research Station, both technical and administrative, with particular thanks to Kristine Holm, Ann-Kathrin Kroken, Simon Flavell, Julie Eikeland, Stian Morken, Hannah Kristine Storum, Lasse Almås Opheim, Lise Dyrhovden, Karen Anita Kvestad and Ivar Helge Matre. Additionally, we would also like to thank those in at the molecular biology laboratory, Institute of Marine Research, Nordnesgaten, with particular thanks to Bjørghild Seliussen, Anne Grete Sørvik and Laila Unneland.

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
