## [Reviewer comments · Royal Society Open Science]

Review History

RSOS-200811.R0 (Original submission)

Review form: Reviewer 1

Is the manuscript scientifically sound in its present form?

Yes

Are the interpretations and conclusions justified by the results?

Yes

Is the language acceptable?

Yes

Do you have any ethical concerns with this paper?

No

Have you any concerns about statistical analyses in this paper?

No

Recommendation?

Accept with minor revision (please list in comments)

Comments to the Author(s)

In this study the authors investigated the impact of domestication on growth and heart morphology in Atlantic salmon using multiple strains with varying levels of domestication. Growth was positively associated with increased domestication. However, genetic background had no impact on heart morphology. Sexual dimorphism was observed in wild strains, with wild female salmon associated with increased heart morphology measurements. This dimorphism was lost in domesticated strains, suggesting domestication selects against these differences in cardiac morphology. Life history stage significantly impacted width-height residuals, with higher measurements in saltwater individuals, indicating rounder hearts.

Comments:

The authors have given a very good explanation of why an alternative metric was used to investigate heart morphology, and dissection and measurements were blinded. The authors also hypothesise the use of F2 hybrids to study the effect of escapees on the wild population, which should provide valuable insight, though this was not put to good use in this particular study. Solid conclusions were provided for some of the observations, such as the loss of sexual dimorphism in the domesticated strains, though the authors failed to provide rationale for other observed differences.

Major Concerns:

The title suggests differences between the strains will be the main focus of the paper. However, the authors primarily discussed differences in heart morphology associated with sex and life history stage.

No information regarding the genotyping procedure. What identification were the fish genotyped for and how accurate was this?

No real data included for liver measurement. The liver was also missing from the introduction and not really discussed at all.

No discussion of the genetic divergence between the different strains in terms of functional genetic differences. Title suggests some genetics would be discussed. Recommend rewording the title to focus on domesticated vs wild.

It is stated that fork length was used as a measure, but there is no description of how this measurement was taken.

Some of the inferences in the results section are not clearly derived from the data presented.

Minor Concerns:

Sentence errors (30-32, 50, 141-142, 153, 317, 354).

Tables on page 34 and 36 don't fit properly.

Weight should be mass.

Reference that AHW was used rather than RHM in the methods section as well as the discussion. "Common garden environment" and "family nested in strain" should be defined in the methods.

Why was wet mass used rather than dry mass?

Interval before hearts were measured is not stated in the methods. Prolonged storage can impact heart morphology.

Review form: Reviewer 2**Is the manuscript scientifically sound in its present form?**

Yes

Are the interpretations and conclusions justified by the results?

Yes

Is the language acceptable?

Yes

Do you have any ethical concerns with this paper?

No

Have you any concerns about statistical analyses in this paper?

No

Recommendation?

Major revision is needed (please make suggestions in comments)

Comments to the Author(s)

See attached (Appendix A).

Review form: Reviewer 3

Is the manuscript scientifically sound in its present form?

Yes

Are the interpretations and conclusions justified by the results?

Yes

Is the language acceptable?

Yes

Do you have any ethical concerns with this paper?

No

Have you any concerns about statistical analyses in this paper?

Yes

Recommendation?

Major revision is needed (please make suggestions in comments)

Comments to the Author(s)

Dear editor

Please find enclosed my comments to the MS entitled « Disentangling the effects of environment and genetics in Atlantic salmon : growth, heart and liver under common garden conditions » by Perry William et al.

General comment

The present study describes the results obtained after a two-year common garden experiment on seven strains of Atlantic salmon (*Salmo salar*) with different number of generations in captivity (up to 13). Three main traits were analysed on more than 2000 fish: growth, heart and liver. Significant differences were only observed for growth; no differences for liver or heart. The introduction is clear, yet I made some suggestions below concerning the use of more recent references. Also, I think that the rationale behind studying the heart and liver is not enough developed; for instance for heart, were any malformations ever observed between wild and

domesticated salmon? Why would you think that either the liver or heart would be modified during domestication? In the materials and methods, why only a subset of salmon were used for certain analyses (L121); no information is given about the tanks used (L122) or any control of water quality (pH, dissolved oxygen, nitrogenous components...); which diet was given to fish (L126). According to this part, it would imply that all fish were reared together within the same two tanks; how they were identified (pit-tag for instance); don't you think that competitions could have occurred between salmon; stress should also have been different; we do expect that stress (any measures of cortisol / different behaviours observed?) decreased as the number of generations in captivity increases? How this could influence your results; particularly concerning growth? L135; both weight and size were measured; why you did not calculate the Fulton's condition index to compare between strains? Why did you use the fork length and not the total length? In the results, as mentioned earlier, don't you think that feeding could have modified growth? How can you be sure that all fish consumed the same amount of feed? How do you explain the difference in growth? Did you observe any mortality (might be different between strains?). For heart, do you observe any malformations? Throughout the MS; you only use the term "domesticated" but you can also use the term "selected" for the strain which is the result of 13 generations of domestication; how many generations were selected for growth? Some information would be interesting to discuss your results. L332; do you have any idea of the IGS for females and males that might strengthen your remark here? Higher IGS values for females might require more energy for instance. In conclusion, I found the MS interesting but quite hard to follow, particularly because some tables lack caption and are not visible on the pdf. If all those changes are made or taken into account, I suggest that a revised manuscript could be re-evaluated.

Specific comments

Abstract

L20: the onset of domestication is much older, if you do consider wolf/dog

L23: it is unclear what you mean by "domesticated environments"?

L31: what do you mean "domesticated line"?

L35-L36: seems to be a pleonasm written like that; be clearer

Introduction

L46: add "animal" before "domestication" (there is also plant domestication)

L54: delete "to domestication", after "particularly salmonids"

L55: might add "effects of" before "on the early stages"

L56-57: I think this sentence is useless here; might keep it for the discussion?

L46-54: I think that other references should be used or added in the first part of the introduction; I may suggest: Teletchea F (2019) Fish domestication in aquaculture: reassessment and emerging questions. *Cybium* 43: 7-15. Teletchea F (2016) Is fish domestication going too fast? *Natural Resources* 7: 399-404. Teletchea F (2015) Domestication and genetics: what a comparison between land and aquatic species can bring? In: Pontarotti P. (eds) *Evolutionary Biology: Biodiversification from genotype to phenotype*, Chapter 20. Springer, pp 389-401. Tave D, Hutson AM (2019) Is Good Fish Culture Management Harming Recovery Efforts in Aquaculture-Assisted Fisheries? *North American Journal of Aquaculture* 81: 333-339. Nakajima T, Hudson MJ, Uchiyama J, Makibayashi K, Zhang J. Common carp aquaculture in Neolithic China dates back 8,000 years. *Nature Ecology and Evolution*.

Line 75: I think that you can find a much more recent reference (20 years old).

L94-95: explain what those differences are; any hypotheses proposed for explaining those differences?

Materials and Methods

L115: might replace "established" by "produced"

L121: delete "With a" and start "A subset of wild [...] "was used"

L123: are you sure about the use of "terminated"; use three times (L125, L128)

L172-173: why measurements were not realized for all fish?

L198: I have no idea how you can calculate 25 or 75% of domestication? What does it mean 100% domestication; if you compare for instance common carp and salmon?

L288: replace "the proportion of domesticated line" by "domestication / selection / number of generations of domestication"

L304: again even though I understand by what you call "0-25-50-75-100%", I suggest using something else

L362: could you add which studies you talk about here

Figure 2

I do not like your X-axis "percentage of domestication"; what does mean 100% of domestication or even 25-25-75... This has no sense for me even though I "understand" what you want to imply by that. I propose that you delete that and replace by number of generations in captivity or of selection, only indicate increasing domestication/selection. Also, I do not understand how the linear regression was performed? What values were used?

Table 1

This table is very important to understand the different strains studied. Yet it is hard to follow. Please modify it to make sure that we can see everything written and add a clear caption.

Supplementary table

This table is not visible in my pdf. Please modify and add a clear caption so it can be understand alone.

Decision letter (RSOS-200811.R0)

Dear Mr Perry,

The editors assigned to your paper ("Disentangling the effects of environment and genetics in Atlantic salmon: growth, heart and liver under common garden conditions") have now received comments from reviewers. We would like you to revise your paper in accordance with the referee and Associate Editor suggestions which can be found below (not including confidential reports to the Editor). Please note this decision does not guarantee eventual acceptance.

Please submit a copy of your revised paper before 09-Jul-2020. Please note that the revision deadline will expire at 00.00am on this date. If we do not hear from you within this time then it will be assumed that the paper has been withdrawn. In exceptional circumstances, extensions may be possible if agreed with the Editorial Office in advance. We do not allow multiple rounds of revision so we urge you to make every effort to fully address all of the comments at this stage. If deemed necessary by the Editors, your manuscript will be sent back to one or more of the original reviewers for assessment. If the original reviewers are not available, we may invite new reviewers.

When submitting your revised manuscript, you must respond to the comments made by the referees and upload a file "Response to Referees" in "Section 6 - File Upload". Please use this to document how you have responded to the comments, and the adjustments you have made. In

order to expedite the processing of the revised manuscript, please be as specific as possible in your response.

- Data accessibility

If you wish to submit your supporting data or code to Dryad (<http://datadryad.org/>), or modify your current submission to dryad, please use the following link:
<http://datadryad.org/submit?journalID=RSOS&manu=RSOS-200811>

- Competing interests

- Authors' contributions

- Acknowledgements

- Funding statement

Kind regards,
Lianne Parkhouse
Editorial Coordinator
Royal Society Open Science
openscience@royalsociety.org

on behalf of Dr Michael Tobler (Associate Editor) and Kevin Padian (Subject Editor)
openscience@royalsociety.org

Associate Editor's comments (Dr Michael Tobler):

We have received the feedback from three reviewers, all of which highlighted the merits of the manuscript. Nonetheless, the reviewers provided constructive feedback in terms of improving analyses, interpretation, and presentation that the authors should address. A rigorously revised manuscript should be acceptable for publication in RSOS.

Reviewers' Comments to Author:

Reviewer: 1
Comments to the Author(s)

In this study the authors investigated the impact of domestication on growth and heart morphology in Atlantic salmon using multiple strains with varying levels of domestication. Growth was positively associated with increased domestication. However, genetic background had no impact on heart morphology. Sexual dimorphism was observed in wild strains, with wild female salmon associated with increased heart morphology measurements. This dimorphism was lost in domesticated strains, suggesting domestication selects against these differences in cardiac morphology. Life history stage significantly impacted width-height residuals, with higher measurements in saltwater individuals, indicating rounder hearts.

Comments:

The authors have given a very good explanation of why an alternative metric was used to investigate heart morphology, and dissection and measurements were blinded. The authors also hypothesise the use of F2 hybrids to study the effect of escapees on the wild population, which should provide valuable insight, though this was not put to good use in this particular study. Solid conclusions were provided for some of the observations, such as the loss of sexual dimorphism in the domesticated strains, though the authors failed to provide rationale for other observed differences.

Major Concerns:

The title suggests differences between the strains will be the main focus of the paper. However, the authors primarily discussed differences in heart morphology associated with sex and life history stage.

No information regarding the genotyping procedure. What identification were the fish genotyped for and how accurate was this?

No real data included for liver measurement. The liver was also missing from the introduction and not really discussed at all.

No discussion of the genetic divergence between the different strains in terms of functional genetic differences. Title suggests some genetics would be discussed. Recommend rewording the title to focus on domesticated vs wild.

It is stated that fork length was used as a measure, but there is no description of how this measurement was taken.

Some of the inferences in the results section are not clearly derived from the data presented.

Minor Concerns:

Sentence errors (30-32, 50, 141-142, 153, 317, 354).

Tables on page 34 and 36 don't fit properly.

Weight should be mass.

Reference that AHW was used rather than RHM in the methods section as well as the discussion.

"Common garden environment" and "family nested in strain" should be defined in the methods.

Why was wet mass used rather than dry mass?

Interval before hearts were measured is not stated in the methods. Prolonged storage can impact heart morphology.

Reviewer: 2

Comments to the Author(s)

See attached file

Reviewer: 3

Comments to the Author(s)

Dear editor

Please find enclosed my comments to the MS entitled « Disentangling the effects of environment and genetics in Atlantic salmon : growth, heart and liver under common garden conditions » by Perry William et al.

General comment

The present study describes the results obtained after a two-year common garden experiment on seven strains of Atlantic salmon (*Salmo salar*) with different number of generations in captivity (up to 13). Three main traits were analysed on more than 2000 fish: growth, heart and liver. Significant differences were only observed for growth; no differences for liver or heart. The introduction is clear, yet I made some suggestions below concerning the use of more recent references. Also, I think that the rationale behind studying the heart and liver is not enough developed; for instance for heart, were any malformations ever observed between wild and domesticated salmon? Why would you think that either the liver or heart would be modify during domestication? In the materials and methods, why only a subset of salmon were used for certain analyses (L121); no information is given about the tanks used (L122) or any control of water quality (pH, dissolved oxygen, nitrogenous components...); which diet was given to fish (L126). According to this part, it would imply that all fish were reared together within the same two tanks; how they were identified (pit-tag for instance); don't you think that competitions could have occurred between salmons; stress should also have been different; we do expect that stress (any measures of cortisol / different behaviours observed?) decreased as the number of generations in captivity increases? How this could influence your results; particularly concerning growth? L135; both weight and size were measured; why you did not calculate the Fulton's condition index to compare between strains? Why did you use the fork length and not the total length? In the results, as mentioned earlier, don't you think that feeding could have modified growth? How can you be sure that all fish consumed the same amount of feed? How do you explain the difference in growth? Did you observed any mortality (might be different between strains?). For heart, do you observe any malformations? Throughout the MS; you only use the term "domesticated" but you can also use the term "selected" for the strain which is the result of 13 generations of domestication; how many generations were selected for growth? Some information would be interesting to discuss your results. L332; do you have any idea of the IGS for females and males that might strengthen you remark here? Higher IGS values for females might require more energy for instance. In conclusion, I found the MS interesting but quite hard

to follow, particularly because some tables lack caption and are not visible on the pdf. If all those changes are made or taken into account, I suggest that a revised manuscript could be re-evaluated.

Specific comments

Abstract

L20: the onset of domestication is much older, if you do consider wolf/dog

L23: it is unclear what you mean by "domesticated environments"?

L31: what do you mean "domesticated line"?

L35-L36: seems to be a pleonasm written like that; be clearer

Introduction

L46: add "animal" before "domestication" (there is also plant domestication)

L54: delete "to domestication", after "particularly salmonids"

L55: might add "effects of " before "on the early stages"

L56-57: I think this sentence is useless here; might keep it for the discussion?

L46-54: I think that other references should be used or added in the first part of the introduction; I

may suggest: Teletchea F (2019) Fish domestication in aquaculture: reassessment and emerging

questions. *Cybium* 43: 7-15. Teletchea F (2016) Is fish domestication going too fast? *Natural*

Resources 7: 399-404. Teletchea F (2015) Domestication and genetics: what a comparison between

land and aquatic species can bring? In: Pontarotti P. (eds) *Evolutionary Biology:*

Biodiversification from genotype to phenotype, Chapter 20. Springer, pp 389-401. Tave D, Hutson

AM (2019) Is Good Fish Culture Management Harming Recovery Efforts in Aquaculture-Assisted

Fisheries? *North American Journal of Aquaculture* 81: 333-339. Nakajima T, Hudson MJ,

Uchiyama J, Makibayashi K, Zhang J. Common carp aquaculture in Neolithic China dates back

8,000 years. *Nature Ecology and Evolution*.

Line 75: I think that you can find a much recent reference (20 years old).

L94-95: explain what those differences are; any hypotheses proposed for explaining those differences?

Materials and Methods

L115: might replace "established" by "produced"

L121: delete "With a" and start "A subset of wild [...] "was used"

L123: are sure about the use of "terminated"; use three times (L125, L128)

L172-173: why measurements were not realized for all fish?

L198: I have no idea how you can calculate 25 or 75% of domestication? What does it mean 100% domestication; if you compare for instance common carp and salmon?

L288: replace "the proportion of domesticated line" by "domestication / selection / number of generations of domestication"

L304: again even though I understand by what you call "0-25-50-75-100%", I suggest using something else

L362: could you add which studies you talk about here

Figure 2

I do not like your X-axis "percentage of domestication"; what does mean 100% of domestication or even 25-25-75... This has no sense for me even though I "understand" what you want to imply by that. I propose that you delete that and replace by number of generations in captivity or of selection, only indicate increasing domestication/selection. Also, I do not understand how the linear regression was performed? What values were used?

Table 1

This table is very important to understand the different strains studied. Yet it is hard to follow. Please modify it to make sure that we can see everything written and add a clear caption.

Supplementary table

This table is not visible in my pdf. Please modify and add a clear caption so it can be understood alone.

Author's Response to Decision Letter for (RSOS-200811.R0)

See Appendix B.

RSOS-200811.R1 (Revision)

Review form: Reviewer 3

Is the manuscript scientifically sound in its present form?

Yes

Are the interpretations and conclusions justified by the results?

Yes

Is the language acceptable?

Yes

Do you have any ethical concerns with this paper?

Yes

Have you any concerns about statistical analyses in this paper?

No

Recommendation?

Accept with minor revision (please list in comments)

Comments to the Author(s)

Dear editor,

The revised manuscript now entitled « Disentangling the effects of sex, life history and genetic background in Atlantic salmon: growth, heart and liver under common garden conditions » has been much improved. I read both the answers to the questions/comments of the three reviewers and the revised MS. I consider that the MS is suitable for publication, even though they are some minor mistakes that should be corrected. Also it was quite impossible to read the tables in the final MS (hopefully some information were much readable in the letter).

Minor comments

Line 33: might add "liver" here after "morphology"?

Lines 47-49: please add the scientific name for "goats", "tilapia" and "carp"

Line 59: add "or liver" after "the heart"?

Line 93: please check this sentence, should be slightly rephrase?

Line 113: add "also" before "fundamental"?

Line 151: how many tanks were used for the saltwater experiment?

Line 257: could you please explain and check how you obtain 2021? Shoud it be 1159 +1043 ?

Line 269: add a figure caption so it can be understood alone

Lines 338-340: check the format of the number after F

The Figure 3b and 3d are not cited in the text.

Table caption

Table 1: add a clear table caption (very hard to read this table in the MS); you can use the same information in the figure caption of figure 2.

Decision letter (RSOS-200811.R1)

Dear Mr Perry

On behalf of the Editors, we are pleased to inform you that your Manuscript RSOS-200811.R1 "Disentangling the effects of sex, life history and genetic background in Atlantic salmon: growth, heart and liver under common garden conditions" has been accepted for publication in Royal Society Open Science subject to minor revision in accordance with the referees' reports. Please find the referees' comments along with any feedback from the Editors below my signature.

Please submit your revised manuscript and required files (see below) no later than 7 days from today's (ie 15-Sep-2020) date. Note: the ScholarOne system will 'lock' if submission of the revision is attempted 7 or more days after the deadline. If you do not think you will be able to meet this deadline please contact the editorial office immediately.

on behalf of Dr Michael Tobler (Associate Editor) and Kevin Padian (Subject Editor)
openscience@royalsociety.org

Reviewer comments to Author:
Reviewer: 3

Comments to the Author(s)
Dear editor,

The revised manuscript now entitled « Disentangling the effects of sex, life history and genetic background in Atlantic salmon: growth, heart and liver under common garden conditions » has been much improved. I read both the answers to the questions/comments of the three reviewers and the revised MS. I consider that the MS is suitable for publication, even though they are some minor mistakes that should be corrected. Also it was quite impossible to read the tables in the final MS (hopefully some information were much readable in the letter).

Minor comments

Line 33: might add "liver" here after "morphology"?

Lines 47-49: please add the scientific name for "goats", "tilapia" and "carp"

Line 59: add "or liver" after "the heart"?

Line 93: please check this sentence, should be slightly rephrase?

Line 113: add "also" before "fundamental"?

Line 151: how many tanks were used for the saltwater experiment?

Line 257: could you please explain and check how you obtain 2021? Shoud it be 1159 +1043 ?

Line 269: add a figure caption so it can be understood alone

Lines 338-340: check the format of the number after F

The Figure 3b and 3d are not cited in the text.

Table caption

Table 1: add a clear table caption (very hard to read this table in the MS); you can use the same information in the figure caption of figure 2.

===PREPARING YOUR MANUSCRIPT===

===PREPARING YOUR REVISION IN SCHOLARONE===

Author's Response to Decision Letter for (RSOS-200811.R1)

See Appendix C.

Decision letter (RSOS-200811.R2)

Dear Mr Perry,

It is a pleasure to accept your manuscript entitled "Disentangling the effects of sex, life history and genetic background in Atlantic salmon: growth, heart and liver under common garden conditions" in its current form for publication in Royal Society Open Science.

Please note that the following email address of one of your co-authors is currently marked as invalid by our system. Please reply to this email with an up-to-date email address for this author:

- afu206@bangor.ac.uk

Best regards,

on behalf of the Associate Editor and Professor Kevin Padian (Subject Editor)
openscience@royalsociety.org

Appendix A

Review RSOS-200811

The study investigates an interesting question that has received little attention to date. Using Atlantic salmon, the study explores how domestication affects internal organs, including the heart and liver, as well as growth. The study is conducted under common-garden conditions and uses various crosses. The crosses conducted include pure wild and farm, as well as F1, F2, and reciprocal backcrosses, thus allowing the authors to investigate how different proportion of domestic genetic background (0 to 100%) influence the traits of interest. The study finds that the proportion of domestic background was associated with growth, where wild fish grow slower (inferred from body size) than domestic fish. No effects of domestication were found on liver or heart characteristics. While no effect of domestication was found, some effects of life stage and sex were identified, with some evidence of sexual dimorphism in heart morphology, including the loss of sexual dimorphisms following domestication.

This study is interesting, and I have no concerns about the methodology or statistical analyses. The sample size included 1159 and 1043 fish at each life stage, respectively, which should be adequate to detect an effect, although it is not clear how many adults contributed to these crosses. A few other details are lacking the Methods (e.g., genotyping methods). The Introduction could use some improvements (as addressed below). As noted below, the liver is not discussed in the Introduction or the Discussion, and this should be addressed even though it was not significant. The Methods are clear although some details are lacking that need to be incorporated. The Results are bit difficult to follow at times, but this is mostly due to many different comparisons that were conducted and the various metrics/acronyms. A Table to summarize the results would be helpful (see comment below). The Discussion provides a good summary of the results, although, I would suggest some re-organization. With some changes, I think this manuscript would be appropriate for publication in RSOS.

Specific comments:

Introduction - The introduction reads relatively well, but I think it could benefit from some re-organization (see comments below). Also, there is no discussion about the liver, but extensive discussion about the heart. More information about why the liver is being studied should be provided in the Introduction. In addition, it is not totally clear why we expect the heart and liver to change with domestication. I think explicitly stated how the aquaculture environment might result in changes would be beneficial here.

Line 52 – “is little studied” – perhaps change to ‘has received little attention’. I would also change ‘though with exception’ to be in parentheses as “(but see [8,9])”

Line 53 – change “looking at” to “investigating”

Line 54 – in “particular” salmonids

Line 92-102 – This paragraph does not seem to fit here. It disrupts the flow of the Introduction. It should be moved prior to the previous paragraph (before Line 81). For example, Line 71 is discussing heart morphology, so perhaps combining with paragraph/section would help. Lines 81-91 should be the last paragraph of the Introduction. The last sentence (Lines 99-102) could be added to the final paragraph.

Lines 99-102 – Isn't liver also examined?

Line 106 -127 – There are no details about the wild population? And how many adults (parents) contributed to the crosses?

Line 121-122 – This is not a complete sentence.

Line 122-123 – All fish were mixed in a tank. Were fish tagged to identify cross/family? I see later this is addressed using DNA. Perhaps indicate here.

Line 136-137 – How was DNA-family identification done? There should be a section added on genetic analyses.

Line 176-177 – Why were these crosses excluded?

Line 193 – 195 – It would be helpful if the selected model could be provided in a Table to easily show the fixed and random factors that were important in the model. Also, it would be helpful for the Table to include p-values for the factors (or at least significant ones) so it is easy to see what factors were significant.

Line 194 – For sex, I didn't notice it mentioned in Methods. Was it done genetically or through internal anatomy? If done using genetics, then what primers were used? This should be added to a genetics section in the text.

Line 212-216 - This is confusing. What are the pairwise comparisons that are significant? There are 4-3 strains discussed, and I'm not sure if that means they were all significantly different from each other or if it's referring to certain comparisons. Re-word for clarity. Also, I don't think MF and FM has been described in the text. I only see it in Figure legend.

Line 224 – This seems a bit short. Perhaps a figure could at least be added to the Supplement to show the lack of differences for the reader.

Line 248 – The "difference" between

Line 284-293 – Again, liver is not mentioned here. It should be discussed to some extent.

Line 308-311 – Because of recombination in the F2 generation, I would expect mean growth to be similar, but I would also expect greater variation in the F2 generation (due to different

potential growth-related genotype combination in this generation). Assuming F1s will be primarily heterozygous for growth loci, F2s could show a range of genotypes for important (heterozygous or homozygous). Was this increase in variance observed?

Line 354 – ...fitted 'using' RHM as .. .

Line 347-363 – This investigation of RHM is interesting, but I wonder if it would fit better in the Methods/Results. It seems a bit out of place to present new analyses in the Discussion. The discussion of the differences between RHM and AH weight certainly make sense here, but it may help to add some details earlier in the paper.

Line 368- males take part in “activities” that require...

Line 405 – This is a good point. It's possible that changes in cardiac morphology could occur later during sexual maturation, when strenuous upstream migrations, mating competition, nest building, etc. will need to take place. It would be helpful to indicate something to that effect here. Sampling sexual mature fish could provide different outcomes.

Line 413 – Move “however” to the start of the sentence.

Figure 2 – Great figure!

Figure 3 – It's difficult to see the marginal means and confidence intervals in these plots. They should be increased in size and shown in black colour.

Figure 4 – Why are there fewer data points here? Indicate more information in figure legend. Significant differences should be indicated.

Tables aren't clear in the PDF. There is a formatting issue here. Cannot see all the text in the table, and no table legend provided.

Rscript – I briefly looked at the R script, and it appears to be clear enough to a person familiar with R, but should be updated with the new metadata.csv provided. There are some variables that do not exist in the data file (which were not used in the study - but are present in the R script; eg. Deformity). I do not think this influences the interpretation of data, but R script should be cleaned up so that it can run without error.

Appendix B

Associate Editor's comments (Dr Michael Tobler):

We have received the feedback from three reviewers, all of which highlighted the merits of the manuscript. Nonetheless, the reviewers provided constructive feedback in terms of improving analyses, interpretation, and presentation that the authors should address. A rigorously revised manuscript should be acceptable for publication in RSOS.

Thank you for the constructive advice. I have made considerable additions to the manuscript (in the form of additional information, subheadings, and tables), as well as the supplementary materials, that address the criticism of the reviewers below.

Reviewers' Comments to Author:

Reviewer: 1

Comments to the Author(s)

In this study the authors investigated the impact of domestication on growth and heart morphology in Atlantic salmon using multiple strains with varying levels of domestication. Growth was positively associated with increased domestication. However, genetic background had no impact on heart morphology. Sexual dimorphism was observed in wild strains, with wild female salmon associated with increased heart morphology measurements. This dimorphism was lost in domesticated strains, suggesting domestication selects against these differences in cardiac morphology. Life history stage significantly impacted width-height residuals, with higher measurements in saltwater individuals, indicating rounder hearts.

Comments:

The authors have given a very good explanation of why an alternative metric was used to investigate heart morphology, and dissection and measurements were blinded. The authors also hypothesise the use of F2 hybrids to study the effect of escapees on the wild population, which should provide valuable insight, though this was not put to good use in this particular study.

Solid conclusions were provided for some of the observations, such as the loss of sexual dimorphism in the domesticated strains, though the authors failed to provide rationale for other observed differences.

Many thanks for the constructive feedback provided here. I hope that the additions and clarification in parts help remedy both the major and minor concerns.

Major Concerns:

The title suggests differences between the strains will be the main focus of the paper. However, the authors primarily discussed differences in heart morphology associated with sex and life history stage.

Title changed to: "Disentangling the effects of **sex, life history** and genetics in Atlantic salmon: growth, heart and liver under common garden conditions"

No information regarding the genotyping procedure. What identification were the fish genotyped for and how accurate was this?

Apologies, this was an oversight. Please find addition of: "and later identified using a genetic parentage analysis" on line 110, and the addition of section "Parentage analysis & genetic sex" on line 135:

"Parentage analysis & genetic sex

Genomic DNA was extracted from alcohol-preserved fin-clip samples using the Qiagen DNeasy@96 Blood & Tissue Kit, followed by a multiplex PCR which amplified six microsatellite loci; SsaF43 [GenBank:U37494] (Sánchez et al., 1996), Ssa197 [GenBank:U43694.1] (O'Reilly

et al., 1996), SSsp3016 [GenBank:AY372820], MHC I (Grimholt et al., 2002) and MHC II (Stet et al., 2002). Genetic sex was identified by the presence of the sdY gene (Yano et al., 2012; Eisbrenner et al., 2014), if the presence of exon 2 and 4 were detected, the individual was designated as male. An ABI Applied Biosystems ABI 3730 Genetic Analyser was used for fragment analysis, the outputs of which were used to call genotypes in GeneMapper (Applied Biosystems, v. 4.0). Further details are outlined by Solberg et al. (2013).”

No real data included for liver measurement. The liver was also missing from the introduction and not really discussed at all.

This is a good point, and the main reason for this is that we did not find any interesting trends regarding the liver, between genetic backgrounds or sexes, and so was not our main focus; however, wanted to still include this to inform future research.

The liver is mentioned briefly in the introduction (line 64), however, we now have the addition of “and liver weight (Poppe et al., 2003; Gamperl and Farrell, 2004; Mayer et al., 2011)” on line 94. Addition of “and liver weight” on line 100.

Addition of “Likewise, farmed fish such as cod (*Gadus morhua*) have been shown to be heavier livers than their wild counterparts, with the development of fatty deposits [11,35]” Line 86

Addition of “Liver” section in the discussion, line 331:

“Liver

We saw no significant effects of life stage, sex, strain or any of the interaction terms on body weight adjusted liver weight (ALW). Previous studies have shown that farmed cod (*Gadus morhua*) have significantly heavier livers than wild counterparts, although this is most likely due to environmental plasticity (Mayer et al., 2011) and the development of enlarged fatty livers when cod are fed high energy lipid-rich diets (Grant, Arown and Boyce, 1998). Despite no significant differences in ALW between wild and domesticated fish found here, there are studies that control for environmental variation which show that gene expression in the liver of salmonids has been impacted by domestication (Normandeau et al., 2009; Tymchuk, Sakhrani and Devlin, 2009). Therefore, it is possible that there could be fine scale histological differences caused by domestication that are not detected by looking at weight alone, providing an avenue for further study.”.

Added. “There were no significant effects of life stage, sex, strain or any of the interaction terms on body weight adjusted liver weight (ALW) (**supplementary figure 2**)”:

Supplementary figure 2 Body weight adjusted liver weight between the seven experimental strains. No significant differences were found between groups in the linear mixed effect models.

No discussion of the genetic divergence between the different strains in terms of functional genetic differences. Title suggests some genetics would be discussed. Recommend rewording the title to focus on domesticated vs wild.

Additional title change to: “Disentangling the effects of sex, life history **and genetic background** in Atlantic salmon: growth, heart and liver under common garden conditions”

It is stated that fork length was used as a measure, but there is no description of how this measurement was taken.

Addition of: “In addition to body weight, fork length (to the nearest 1 mm) was also measured using a standard measuring board.” Line 146.

Some of the inferences in the results section are not clearly derived from the data presented.

Maybe the lack of legible tables contributed to this? Following on from this comment, I have also moved table 2 into the supplementary materials, as I do not think that it contributed to the discussion and confused the narrative.

Minor Concerns:
Sentence errors (30-32,

Tried to clarify by removing some of the commas, if this is what was meant by sentence error?

“Growth was positively linked with the proportion of the domesticated line, and recombination in F2 hybrids (and the potential disruption of co-adapted gene complexes) did not influence growth”

50,

Broken sentence down into two, and clarified. “Studies examining the impacts of domestication have typically focused on features such as bone shape [5], behaviour [6] and genetics [7].

Assessment of internal organs in domesticated taxa, including structural heart morphology, are little studied, though with exceptions [8,9].”

141-142,

Repeated sentence removed.

153,

Fixed.

317,

Removal of repetition of immature smolt.

354).

Fixed.

Tables on page 34 and 36 don't fit properly.

I think this is a problem with how the RSOS website compiles all of the files. Please find a better formatted table here:

Table 1 on page 34:

Origin			wet weight		heart weight		heart morphology (AHH, AHW, WHR)	
type	strain	location	freshwater	saltwater	freshwater	saltwater	freshwater	saltwater
domesticated	Mowi		♂ = 98 ♀ = 80	♂ = 72 ♀ = 93	♂ = 43 ♀ = 48	♂ = 34 ♀ = 42	♂ = 8 ♀ = 15	♂ = 12 ♀ = 15
hybrid	Figgjo (♀) × Mowi (♂)		♂ = 59 ♀ = 38	♂ = 52 ♀ = 32	♂ = 34 ♀ = 24	♂ = 24 ♀ = 16	♂ = 14 ♀ = 1	♂ = 13 ♀ = 9
	Mowi (♀) × Figgjo (♂)		♂ = 57 ♀ = 39	♂ = 53 ♀ = 43	♂ = 23 ♀ = 23	♂ = 16 ♀ = 16	♂ = 5 ♀ = 3	♂ = 6 ♀ = 8
F2 hybrid			♂ = 95 ♀ = 76	♂ = 92 ♀ = 98	♂ = 50 ♀ = 43	♂ = 45 ♀ = 43		
wild	Figgjo	58°81' N, 5°55' E	♂ = 103 ♀ = 81	♂ = 72 ♀ = 76	♂ = 54 ♀ = 52	♂ = 29 ♀ = 28	♂ = 13 ♀ = 12	♂ = 10 ♀ = 13
wild backcross			♂ = 90 ♀ = 78	♂ = 86 ♀ = 87	♂ = 36 ♀ = 41	♂ = 33 ♀ = 38		
domesticated backcross			♂ = 84 ♀ = 78	♂ = 92 ♀ = 79	♂ = 47 ♀ = 44	♂ = 45 ♀ = 30		

Table 2 on page 36: (NOW IN SUPPLEMENTARY MATERIALS)

contrast	freshwater	saltwater	trend between strains
Wild - Wild.BC	0.2403	0.0193	increase
Wild - Hybrid.FM	0.0004	<.0001	increase
Wild - Hybrid.MF	0.0001	<.0001	increase
Wild - F2	<.0001	<.0001	increase
Wild - Domesticated.BC	<.0001	<.0001	increase
Wild - Domesticated	<.0001	<.0001	increase
Wild.BC - Hybrid.FM	0.158	0.2095	increase
Wild.BC - Hybrid.MF	0.0723	0.001	increase
Wild.BC - F2	0.0795	0.0022	increase
Wild.BC - Domesticated.BC	0.0003	<.0001	increase
Wild.BC - Domesticated	<.0001	<.0001	increase
Hybrid.FM - Hybrid.MF	1	0.8393	increase
Hybrid.FM - F2	1	0.9992	increase
Hybrid.FM - Domesticated.BC	0.9686	0.2008	increase
Hybrid.FM - Domesticated	0.0051	0.0007	increase
Hybrid.MF - F2	1	0.9946	decrease
Hybrid.MF - Domesticated.BC	0.9974	0.9998	increase
Hybrid.MF - Domesticated	0.0134	0.1651	increase
F2 - Domesticated.BC	0.6853	0.4591	increase
F2 - Domesticated	0.0001	0.0008	increase
Domesticated.BC - Domesticated	0.0385	0.3205	increase

Weight should be mass.

This is an interesting point. We used weighing scales, which measure the amount of force that object puts on the scale, giving us the weight. Mass is measured by using a balance comparing a known amount of matter to an unknown amount of matter. Therefore, we will stick with weight, as this is what we measured, as well as to keep it consistent with the literature.

Reference that AHW was used rather than RHM in the methods section as well as the discussion.

Added: "AH weight was used instead of relative heart mass." Line 185.

Added: "(measure used instead of relative heart mass (RHM))" Line 356

"Common garden environment" and "family nested in strain" should be defined in the methods.

Addition: "A common garden environment ensures that all experimental strains are reared under the same environmental conditions by rearing them together"

Addition: "Family was nested within strain in order to account for non-independence in the data, as families are unique to their corresponding strain."

Why was wet mass used rather than dry mass?

As we did not have the capacity to dry every fish.

Interval before hearts were measured is not stated in the methods. Prolonged storage can impact heart morphology.

This is certainly an issue, however, as all of the hearts between sex and genetic background had all spent the same amount of time stored in formalin, any storage effect should be consistent across experimental groups.

Reviewer: 2

Comments to the Author(s)

The study investigates an interesting question that has received little attention to date. Using Atlantic salmon, the study explores how domestication affects internal organs, including the heart and liver, as well as growth. The study is conducted under common-garden conditions and uses various crosses. The crosses conducted include pure wild and farm, as well as F1, F2, and reciprocal backcrosses, thus allowing the authors to investigate how different proportion of domestic genetic background (0 to 100%) influence the traits of interest. The study finds that the proportion of domestic background was associated with growth, where wild fish grow slower (inferred from body size) than domestic fish. No effects of domestication were found on liver or heart characteristics. While no effect of domestication was found, some effects of life stage and sex were identified, with some evidence of sexual dimorphism in heart morphology, including the loss of sexual dimorphisms following domestication.

This study is interesting, and I have no concerns about the methodology or statistical analyses. The sample size included 1159 and 1043 fish at each life stage, respectively, which should be adequate to detect an effect, **although it is not clear how many adults contributed to these crosses**. A few other details are lacking **the Methods (e.g., genotyping methods)**.

Please see addition of “Parentage analysis & genetic sex” section:

“Parentage analysis & genetic sex

Genomic DNA was extracted from alcohol-preserved fin-clip samples using the Qiagen DNeasy®96 Blood & Tissue Kit, followed by a multiplex PCR which amplified seven microsatellite loci; SsaF43 [GenBank:U37494] (Sánchez et al., 1996), Ssa197 [GenBank:U43694.1] (O’Reilly et al., 1996), SSsp3016 [GenBank:AY372820], MHCI (Grimholt et al., 2002), MHCII (Stet et al., 2002), as well as the sex markers exon 2 and exon 4. An ABI Applied Biosystems ABI 3730 Genetic Analyser was used for fragment analysis, the outputs of which were used to call genotypes in GeneMapper (Applied Biosystems, v. 4.0). Further details are outlined by Solberg et al. (2013).”

The Introduction could use some improvements (as addressed below). **As noted below, the liver is not discussed in the Introduction or the Discussion, and this should be addressed even though it was not significant.**

This is a good point, we now have the addition of “and liver weight (Poppe et al., 2003; Gamperl and Farrell, 2004; Mayer et al., 2011)” on line 94. Addition of “and liver weight” on line 100.

Addition of “Likewise, farmed fish such as cod (*Gadus morhua*) have been shown to be heavier livers than their wild counterparts, with the development of fatty deposits [11,35]” Line 86

Addition of “Liver” section in the discussion, line 331:

“Liver

We saw no significant effects of life stage, sex, strain or any of the interaction terms on body weight adjusted liver weight (ALW). Previous studies have shown that farmed cod (*Gadus morhua*) have significantly heavier livers than wild counterparts, although this is most likely due to environmental plasticity (Mayer et al., 2011) and the development of enlarged fatty livers when cod are fed high energy lipid-rich diets (Grant, Arown and Boyce, 1998). Despite no significant differences in ALW between wild and domesticated fish found here, there are studies that control for environmental variation which show that gene expression in the liver of salmonids has been impacted by domestication (Normandeau et al., 2009; Tymchuk, Sakhrani and Devlin, 2009). Therefore, it is possible that there could be fine scale histological differences caused by domestication that are not detected by looking at weight alone, providing an avenue for further study.”.

The Methods are clear although some details are lacking that need to be incorporated. The Results are bit difficult to follow at times, but this is mostly due to many different comparisons that were conducted and the various metrics/acronyms. A Table to summarize the results would be helpful (see comment below). The Discussion provides a good summary of the results, although, **I would suggest some re-organization**. With some changes, I think this manuscript would be appropriate for publication in RSOS.

Many thanks for the constructive feedback provided here. I hope that the additions, restructuring, and clarification in parts help remedy the concerns highlighted.

Specific comments:

Introduction - The introduction reads relatively well, but I think it could benefit from some reorganization

(see comments below). Also, there is no discussion about the liver, but extensive discussion about the heart. More information about why the liver is being studied should be provided in the Introduction. In addition, it is not totally clear why we expect the heart and liver to change with domestication. I think explicitly stated how the aquaculture environment might result in changes would be beneficial here.

This is a good point, we now have the addition of “and liver weight (Poppe et al., 2003; Gamperl and Farrell, 2004; Mayer et al., 2011)” on line 94. Addition of “and liver weight” on line 100.

Addition of “Likewise, farmed fish such as cod (*Gadus morhua*) have been shown to be heavier livers than their wild counterparts, with the development of fatty deposits [11,35]” Line 86

Addition of “Liver” section in the discussion:

“Liver

We saw no significant effects of life stage, sex, strain or any of the interaction terms on body weight adjusted liver weight (ALW). Previous studies have shown that farmed cod (*Gadus morhua*) have significantly heavier livers than wild counterparts, although this is most likely due to environmental plasticity (Mayer et al., 2011) and the development of enlarged fatty livers when cod are fed high energy lipid-rich diets (Grant, Arown and Boyce, 1998). Despite no significant differences in ALW between wild and domesticated fish found here, there are studies that control for environmental variation which show that gene expression in the liver of salmonids has been

impacted by domestication (Normandeau et al., 2009; Tymchuk, Sakhrani and Devlin, 2009). Therefore, it is possible that there could be fine scale histological differences caused by domestication that are not detected by looking at weight alone, providing an avenue for further study.”

Line 52 – “is little studied” – perhaps change to ‘has received little attention’. I would also change ‘though with exception’ to be in parentheses as “(but see [8,9])”

Fixed

Line 53 – change “looking at” to “investigating”

Fixed

Line 54 – in “particular” salmonids

Fixed

Line 92-102 – This paragraph does not seem to fit here. It disrupts the flow of the Introduction. It should be moved prior to the previous paragraph (before Line 81). For example, Line 71 is discussing heart morphology, so perhaps combining with paragraph/section would help. Lines 81-91 should be the last paragraph of the Introduction. The last sentence (Lines 99-102) could be added to the final paragraph.

I agree, layout has been changed.

Lines 99-102 – Isn't liver also examined?

Apologies. Liver now added to this sentence.

Line 106 -127 – There are no details about the wild population?

Addition of “from the river Figgjo 58°819N, 5°559E”

And how many adults (parents) contributed to the crosses?

Addition of family information in table 1

Addition of “Seven experimental strains, **represented by 36 different families and produced by 18 male and 18 female parent broodstock, were utilised in this experiment, including**”.
Line 134.

Line 121-122 – This is not a complete sentence.

Fixed.

Line 122-123 – All fish were mixed in a tank. Were fish tagged to identify cross/family? I see

later this is addressed using DNA. Perhaps indicate here.

Added “To identify the genetic background of fish in the common garden post-sampling, a genetic parentage analysis was conducted.”

Line 136-137 – How was DNA-family identification done? There should be a section added on genetic analyses.

Apologies, added section “Parentage analysis”

Line 176-177 – Why were these crosses excluded?

As we did not see any significant differences between wild, domesticated and hybrid fish, we decided not to further measure the backcrosses and F2s, as they would be unlikely to add anything to the dataset.

Line 193 – 195 – It would be helpful if the selected model could be provided in a Table to easily show the fixed and random factors that were important in the model. Also, it would be helpful for the Table to include p-values for the factors (or at least significant ones) so it is easy to see what factors were significant.

Addition of summary section in results, with corresponding table:

“Summary

Strain was only seen as a significant effect in the linear mixed effect model assessing body weight (table 2). In addition to this, family variation was also a significant random effect in all linear mixed effect models, other than the model assessing ALW. Finally, sex was included in all models, apart from the model assessing ALW, and was a significant factor independently, or due to an interaction with another factor, in all instances.”

	fixed factors						random					
	life stage	sex	strain	life stage*sex	life stage*strain	sex*strain	life stage*strain*sex	date of dissection	dam	sire	strain*family	tank
body weight	✓ F1,2 = 2031.39, p < 0.01	✓ F1,2021 = 11.28, p < 0.01	✓ F6,28 = 38.70, p < 0.01	X	✓ F6,2028 = 7.25, p < 0.01	✓ F6,2022 = 5.50, p < 0.01	X	X	X	X	✓ LRT = 204.43, p < 0.01	✓ LRT = 14.03, p < 0.01
adjusted liver weight (ALW)	X	X	X	X	X	X	X	X	X	X	X	X
adjusted heart weight (AH weight)	X	✓ F1,965 = 6.28, p < 0.01	X	X	X	X	X	✓ LRT = 16.52, p < 0.01	X	X	✓ LRT = 46.71, p < 0.01	X
adjusted heart height (AHH)	✓ F1,140 = 5.80, p = 0.02	✓ F1,137 = 0.57, p = 0.45	✓ F3,13 = 1.57, p = 0.85	✓ F1,137 = 7.36, p < 0.01	X	✓ F3,135 = 3.72, p < 0.01	X	X	X	X	✓ LRT = 11.76, p = 0.01	X
adjusted heart width (AH width)	X	✓ F1,165 = 12.76, p < 0.01	X	X	X	X	X	X	X	X	✓ LRT = 6.69, p = 0.01	X
heart width-height residuals (WHR)	✓ F1,142 = 10.51, p < 0.01	✓ F1,143 = 0.27, p = 0.61	X	✓ F1,139 = 4.03, p = 0.05	X	X	X	X	X	X	✓ LRT = 18.62, p < 0.01	X

“Table 3 Linear mixed effect model summary table, assessing the response variables: body weight, body weight adjusted liver weight (ALW), body weight adjusted heart weight (AH weight), fork length adjusted heart height (AHH), for length adjusted heart width (AH width), heart width-height residuals (WHR). The effect of fixed factors: life stage, sex, strain, their interaction terms, as well as random factors: date of dissection, dam, sire, family nested in strain and tank are displayed. Included are factors that contributed to the final model (✓), and those that were removed by the ‘step’ function within ‘lme4’ (X). Significant factors are in bold, with F values and degrees of freedom included for all fixed factors that contributed to the final model, as well as likelihood ratio test (LRT) for random effects.”

Line 194 – For sex, I didn’t notice it mentioned in Methods. Was it done genetically or through internal anatomy? If done using genetics, then what primers were used? This should be added to a genetics section in the text.

Apologies, this should have been included. Addition of: “as well as the sex markers exon 2 and exon 4.” In parentage section. Line 166.

Line 212-216 - This is confusing. What are the pairwise comparisons that are significant? There are 4-3 strains discussed, and I'm not sure if that means they were all significantly different from each other or if it's referring to certain comparisons. Re-word for clarity.

Added numbers and made it more explicit to add clarity:

“Differences between 1) the wild backcross strain and the wild strain, 2) the wild backcross strain and the hybrid FM strain and 3) the wild backcross strain and the F2 strain, went from being non-significant in the fresh water life stage measurement ($P > 0.05$) to significantly different in the saltwater life stage measurement (Table 2).”

Also, I don't think MF and FM has been described in the text. I only see it in Figure legend.

Addition of “hybrid FM (Figgjo (♀) × Mowi (♂)), hybrid MF (Mowi (♀) × Figgjo (♂))”. Line 123

Line 224 – This seems a bit short. Perhaps a figure could at least be added to the Supplement to show the lack of differences for the reader.

Added. “There were no significant effects of life stage, sex, strain or any of the interaction terms on body weight adjusted liver weight (ALW) (**supplementary figure 2**)”:

Supplementary figure 2 Body weight adjusted liver weight between the seven experimental strains. No significant differences were found between groups in the linear mixed effect models.

Line 248 – The “difference” between

Fixed.

Line 284-293 – Again, liver is not mentioned here. It should be discussed to some extent.

Addition of Liver section on line 336.

Line 308-311 – Because of recombination in the F2 generation, I would expect mean growth to be similar, but I would also expect greater variation in the F2 generation (due to different potential growth-related genotype combination in this generation). Assuming F1s will be primarily heterozygous for growth loci, F2s could show a range of genotypes for important (heterozygous or homozygous). Was this increase in variance observed?

This is an very interesting point, and I agree, this is what you would expect to see. Looking at the standard deviation per strain per life stage, however, this is not what we see:

Wild and wild backcross seem to have the largest SD. This could be because we would have to compare the variance of the F2 with it's parental F1's. In the design we have here the F1's are not necessarily related to the F2's.

Addition of: A full breakdown of estimated marginal means per strain and per life stage with standard deviation and standard error can be found in supplementary table 4.

Supplementary table 4 Estimated marginal means (emmean) for log10 wet body weight (g), per strain, broken down by life stage, along with corresponding standard deviation (sd), standard error (SE), degrees of freedom (df) and upper (upper.CL) and lower confidence limits (lower.CL).

Lifestage strain	emmean	sd	SE	df	lower.CL	upper.CL
Freshwater_Domesticated	2.2	0.1536111	0.0349	24.4	2.13	2.27
Freshwater_Domesticated.BC	2.03	0.1348705	0.0351	24.9	1.96	2.1
Freshwater_F2	1.94	0.144485	0.035	24.6	1.86	2.01
Freshwater_Hybrid.FM	1.95	0.1419602	0.0468	32.3	1.86	2.05
Freshwater_Hybrid.MF	1.97	0.1916356	0.0468	32.2	1.88	2.07
Freshwater_Wild	1.66	0.2093108	0.0348	24.1	1.58	1.73
Freshwater_Wild.BC	1.78	0.1853998	0.035	24.7	1.71	1.86
Saltwater_Domesticated	3.22	0.1245083	0.0351	24.9	3.14	3.29
Saltwater_Domesticated.BC	3.09	0.1646113	0.035	24.5	3.02	3.17
Saltwater_F2	2.99	0.17933	0.0347	23.9	2.91	3.06
Saltwater_Hybrid.FM	2.93	0.13775	0.0472	33.4	2.84	3.03
Saltwater_Hybrid.MF	3.05	0.121975	0.0467	32	2.95	3.14
Saltwater_Wild	2.59	0.2320068	0.0353	25.5	2.52	2.67
Saltwater_Wild.BC	2.77	0.1993173	0.035	24.5	2.7	2.84

Line 354 – ...fitted 'using' RHM as ...

Fixed.

Line 347-363 – This investigation of RHM is interesting, but I wonder if it would fit better in the Methods/Results. It seems a bit out of place to present new analyses in the Discussion. The discussion of the differences between RHM and AH weight certainly make sense here, but it may help to add some details earlier in the paper.

I agree. Addition of: “In addition to the analysis of our data, a proof of concept was also included, contrasting the outcome of two different methods in adjusting for allometry: 1) use of residuals from a regression between the measure of interest and a measure of body size, and 2) divisional indexes whereby the measure of interest is divided by a measure of body size. We use the saltwater subset of our data, adjusting heart weight using the two methods, resulting in the residual based AH weight, and the division based relative heart mass (RHM). RHM is calculated using the following formula: heart weight (g) / body weight (g) x 100. Both AH weight and RHM were included as response variables in a linear regression with body weight. To further show the problems of using RHM, neutrally simulated data whereby random simulations (n = 990) (using the range, standard deviation and mean of the observed heart and body weight data) was produced. The randomly produced dataset was then used to calculate a neutrally simulated RHM, which was subsequently used as the response variable in a regression with the neutrally simulated body weight. Finally, using the entire dataset, the same full model for AH weight (as described above) was fitted with RHM as the response variable. After the step function, the final model for RHM contained the fixed factors sex and life stage as well as the random factors family nested in strain and dissection date.” In methods.

Addition of: “Proof of concept: residuals vs division when adjusting for body size

The regression between AH weight and fish weight for the saltwater individuals used in this study was not significant ($R^2 = 0.006$, $F(1,425) = 2.57$, $P = 0.11$); the regression between RHM and fish weight, however, was significant ($R^2 = 0.060$, $F(1,423) = 26.81$, $P < 0.001$) (figure 6 a & b). The regression between naturally simulated RMV and body weight was also significant ($R^2 =$

0.252, $F(1,988) = 334.4$, $P < 0.001$) (figure 6c). When using the entire dataset, a significant effect on RHM was seen for life stage (LME Life stage: $F_{1,15} = 61.83$, $\text{Sum Sq} = 0.050$, $P < 0.001$), with a significant increase in RHM seen in freshwater individuals when compared to saltwater. Sex was also a significant effect (LME Sex: $F_{1,967} = 10.64$, $\text{Sum Sq} = 0.009$, $P < 0.01$), with a significant increase in RHM seen in males when compared to females." In results.

Reworded discussion.

Line 368- males take part in "activities" that require...

Fixed.

Line 405 – This is a good point. It's possible that changes in cardiac morphology could occur later during sexual maturation, when strenuous upstream migrations, mating competition, nest building, etc. will need to take place. It would be helpful to indicate something to that effect here. Sampling sexual mature fish could provide different outcomes.

Addition of: "Spawning multiple times results in multiple upstream migrations, as well as more competing for mates and nest building, which could affect heart morphology. Likewise, even if the fish had not spawned before, they may have spent a greater length of time at sea or may be at different levels of sexual maturation; all of which could also impact on heart morphology."

Line 413 – Move "however" to the start of the sentence.

Fixed.

Figure 2 – Great figure!

Thank you!

Figure 3 – It's difficult to see the marginal means and confidence intervals in these plots. They should be increased in size and shown in black colour.

Fixed.

Figure 4 – Why are there fewer data points here? Indicate more information in figure legend.

This is because adjusted heart height, adjusted heart width and width-height residuals were only performed on a subset of the samples that had data for heart weight. Addition of: "Sample numbers for AHH and WHR between sex, life stage and genetic background are outlined in table 1."

Significant differences should be indicated.

Addition of: Significant ($p < 0.05$) pairwise comparisons are shown with brackets and an asterisk.

Tables aren't clear in the PDF. There is a formatting issue here. Cannot see all the text in the table, and no table legend provided.

I think this is a problem with how the RSOS website compiles all of the files. Please find a better formatted table here:

Table 1 on page 34:

Origin type	strain	number of families	location	wet weight		heart weight		heart morphology (A)	
				freshwater	saltwater	freshwater	saltwater	freshwater	saltwater
domesticated	Mowi	6		♂ = 98 ♀ = 80	♂ = 72 ♀ = 93	♂ = 43 ♀ = 48	♂ = 34 ♀ = 42	♂ = 8 ♀ = 15	♂ = 1
hybrid	Figgjo (♀) × Mowi (♂)	3		♂ = 59 ♀ = 38	♂ = 52 ♀ = 32	♂ = 34 ♀ = 24	♂ = 24 ♀ = 16	♂ = 14 ♀ = 1	♂ = 1
	Mowi (♀) × Figgjo (♂)	3		♂ = 57 ♀ = 39	♂ = 53 ♀ = 43	♂ = 23 ♀ = 23	♂ = 16 ♀ = 16	♂ = 5 ♀ = 3	♂ = 1
F2 hybrid		6		♂ = 95 ♀ = 76	♂ = 92 ♀ = 98	♂ = 50 ♀ = 43	♂ = 45 ♀ = 43		
wild	Figgjo	6	58°81' N, 5°55' E	♂ = 103 ♀ = 81	♂ = 72 ♀ = 76	♂ = 54 ♀ = 52	♂ = 29 ♀ = 28	♂ = 13 ♀ = 12	♂ = 1
wild backcross		6		♂ = 90 ♀ = 78	♂ = 86 ♀ = 87	♂ = 36 ♀ = 41	♂ = 33 ♀ = 38		
domesticated backcross		6		♂ = 84 ♀ = 78	♂ = 92 ♀ = 79	♂ = 47 ♀ = 44	♂ = 45 ♀ = 30		

Table 2 on page 36: (NOW IN SUPPLEMENTARY MATERIALS)

contrast	freshwater	saltwater	trend between strains
Wild - Wild.BC	0.2403	0.0193	increase
Wild - Hybrid.FM	0.0004	<.0001	increase
Wild - Hybrid.MF	0.0001	<.0001	increase
Wild - F2	<.0001	<.0001	increase
Wild - Domesticated.BC	<.0001	<.0001	increase
Wild - Domesticated	<.0001	<.0001	increase
Wild.BC - Hybrid.FM	0.158	0.2095	increase
Wild.BC - Hybrid.MF	0.0723	0.001	increase
Wild.BC - F2	0.0795	0.0022	increase
Wild.BC - Domesticated.BC	0.0003	<.0001	increase
Wild.BC - Domesticated	<.0001	<.0001	increase
Hybrid.FM - Hybrid.MF	1	0.8393	increase
Hybrid.FM - F2	1	0.9992	increase
Hybrid.FM - Domesticated.BC	0.9686	0.2008	increase
Hybrid.FM - Domesticated	0.0051	0.0007	increase
Hybrid.MF - F2	1	0.9946	decrease
Hybrid.MF - Domesticated.BC	0.9974	0.9998	increase
Hybrid.MF - Domesticated	0.0134	0.1651	increase
F2 - Domesticated.BC	0.6853	0.4591	increase
F2 - Domesticated	0.0001	0.0008	increase
Domesticated.BC - Domesticated	0.0385	0.3205	increase

Rscript – I briefly looked at the R script, and it appears to be clear enough to a person familiar with R, but should be updated with the new metadata.csv provided. There are some variables that do not exist in the data file (which were not used in the study - but are present in the R script; eg. Deformity). I do not think this influences the interpretation of data, but R script should be cleaned up so that it can run without error.

Fixed.

Reviewer: 3

Comments to the Author(s)

General comment

The present study describes the results obtained after a two-year common garden experiment on seven strains of Atlantic salmon (*Salmo salar*) with different number of generations in captivity (up to 13). Three main traits were analysed on more than 2000 fish: growth, heart and liver. Significant differences were only observed for growth; no differences for liver or heart. The introduction is clear, yet I made some suggestions below concerning the use of more recent references.

Also, I think that the rationale behind studying the heart and liver is not enough developed; for instance for heart, were any malformations ever observed between wild and domesticated

salmon? Why would you think that either the liver or heart would be modified during domestication?

In the materials and methods, why only a subset of salmon were used for certain analyses (L121);

This was simply due to the fact that measuring the heart width and heart length is far more labour intensive than measuring weight, and so this restricted the number of hearts that could be measured. As for the strains we used: we did not include the backcrosses or F2s for the heart width and heart length, as we did not see any trend between domesticated, wild and F1 hybrids, and therefore, we decided that it was extremely unlikely for any trends to be found with backcrosses and F2s.

No information is given about the tanks used (L122) or any control of water quality (pH, dissolved oxygen, nitrogenous components...); which diet was given to fish (L126).

We did not have any measures of water quality, however, these were flow through tanks, rather than enclosed RAS systems, and so water quality should not have been an issue. I have added a lot more detail on rearing, including diet, please see line 132.

According to this part, it would imply that all fish were reared together within the same two tanks; how they were identified (pit-tag for instance);

Apologies, this information should have been included. They were identified using DNA parentage after sampling. Please see addition of: "To identify the genetic background of fish in the common garden post-sampling, a genetic parentage analysis was conducted." And section "parentage analysis and genetic sex" on line 148.

don't you think that competitions could have occurred between salmon; stress should also have been different; we do expect that stress (any measures of cortisol / different behaviours observed?) decreased as the number of generations in captivity increases? How this could influence your results; particularly concerning growth?

This is a good point, and is the focus of this paper which validates the common garden design approach in wild, farmed, hybrid, salmon:

Solberg, M. F., Zhang, Z., Nilsen, F., & Glover, K. A. (2013). Growth reaction norms of domesticated, wild and hybrid Atlantic salmon families in response to differing social and physical environments. *BMC evolutionary biology*, 13(1), 234.

In short: "there was no difference in relative growth between wild and farmed salmon when reared together and separately".

L135; both weight and size were measured; why you did not calculate the Fulton's condition index to compare between strains?

Much like our argument in the paper about relative heart weight, when you divide heart weight by body weight, Fulton's condition index falls into the same problem. The division increases type I error, and thus false positives. In addition to this, we were not assessing condition, but were instead interested in growth.

Why did you use the fork length and not the total length?

We can more accurately measure fork length on a measuring board, and is not influenced by fin damage like total length might be.

In the results, as mentioned earlier, don't you think that feeding could have modified growth? How can you be sure that all fish consumed the same amount of feed? How do you explain the difference in growth?

An aspect of feeding could definitely be contributing to the differences in growth rate, with a genetically-increased appetite has been suggested by some authors. Addition of: "One

explanation that has been suggested for the increased growth rates seen in domesticated fish is genetically-increased appetite [54].” Line 338

Did you observed any mortality (might be different between strains?). For heart, do you observe any malformations?

We did not calculate this for each of the groups, as mortality is very very low in these tanks. This would be a different story in a wild common garden, however.

Throughout the MS; you only use the term “domesticated” but you can also use the term “selected” for the strain which is the result of 13 generations of domestication; how many generations were selected for growth? Some information would be interesting to discuss your results.

I think from pretty early on, as it was one of the easiest and most effective traits to measure. Addition of: “ which show the effect of strong directional selection placed on growth since the original breeding programs ~13 generations ago” line 323

L332; do you have any idea of the IGS for females and males that might strengthen you remark here? Higher IGS values for females might require more energy for instance.

Are you referring to Gonadosomatic index? GSI? We did not have GSI for these individuals, as they had not yet reached sexual maturity. But I agree, this would have been a useful parameter to test this hypothesis, however, sadly, we do not have this data. I will add, however, that GSI is another one of those divisional indexes that suffer from inflated type I error!

In conclusion, I found the MS interesting but quite hard to follow, particularly because some tables lack caption and are not visible on the pdf. If all those changes are made or taken into account, I suggest that a revised manuscript could re re-evaluated.

Thank you! Apologies about the tables, this appears to be a problem with compilation by the RSOS website.

Specific comments

Abstract

L20: the onset of domestication is much older, if you do consider wolf/dog

Good point. Addition of “livestock”.

L23: it is unclear what you mean by “domesticated environments”?

Replaced with artificial environments.

L31: what to you mean “domesticated line”

Changed to strain.

L35-L36: seems to be a pleonasm written like that; be clearer

Clarified.

Introduction

L46: add “animal” before “domestication” (there is also plant domestication)

Added livestock, to fit with earlier comment about the abstract.

L54: delete “to domestication”, after “particularly salmonids”

Fixed.

L56-57: I think this sentence is useless here; might keep it for the discussion?

I am trying to give a broader background to domestication here, and how it can be used to understand evolutionary principles. I think this broader context is more suited to the introduction than in the discussion.

L46-54: I think that other references should be used or added in the first part of the introduction; I may suggest:

Teletchea F (2019) Fish domestication in aquaculture: reassessment and emerging questions. *Cybium* 43: 7-15.

Added.

Teletchea F (2016) Is fish domestication going too fast? *Natural Resources* 7: 399-404.

Added.

Teletchea F (2015) Domestication and genetics: what a comparison between land and aquatic species can bring? In: Pontarotti P. (eds) *Evolutionary Biology: Biodiversification from genotype to phenotype*, Chapter 20. Springer, pp 389-401.

Added.

Tave D, Hutson AM (2019) Is Good Fish Culture Management Harming Recovery Efforts in Aquaculture-Assisted Fisheries? *North American Journal of Aquaculture* 81: 333-339.

Nakajima T, Hudson MJ, Uchiyama J, Makibayashi K, Zhang J. Common carp aquaculture in Neolithic China dates back 8,000 years. *Nature Ecology and Evolution*.

Added.

Line 75: I think that you can find a much recent reference (20 years old).

Crazy how 2000 is now an old reference! How time flies. But point taken. Updated.

L94-95: explain what those differences are; any hypotheses proposed for explaining those differences?

Addition of: "These studies have revealed differences in heart morphology between farmed and wild fish, with farmed fish displaying more rounded hearts, a hallmark of a more sedentary fish species; along with deposition of fat around the heart. Likewise, farmed fish such as cod (*Gadus morhua*) have been shown to be heavier livers than their wild counterparts, with the development of fatty deposits [11,35]."

Materials and Methods

L115: might replace "established" by "produced"

Fixed.

L121: delete "With a" and start "A subset of wild [...] "was used"

Fixed

L123: are sure about the use of "terminated"; use three time (L125, L128)

Changed to euthanised

L172-173: why measurements were not realized for all fish?

This was simply due to the fact that measuring the heart width and heart length is far more labour intensive than measuring weight, and so this restricted the number of hearts that could be measured. As for the strains we used: we did not include the backcrosses or F2s for the heart width and heart length, as we did not see any trend between domesticated, wild and F1 hybrids, and therefore, we decided that it was extremely unlikely for any trends to be found with backcrosses and F2s.

L198: I have no idea how you can calculate 25 or 75% of domestication? What does it mean 100% domestication; if you compare for instance common carp and salmon?

Addition to figure 2 legend: "Percentage level of domestication used here is calibrated to our experimental design, with our domesticated strain being 100% domesticated, and our wild strain being 0% domesticated. Intermediate levels of domestication are produced through different combinations of hybridisation, be it direct hybrids (50% domesticated, 50% wild), wild backcrosses (25% domesticated, 75% wild) or domesticated backcrosses (75% domesticated, 25% wild).

L288: replace "the proportion of domesticated line" by "domestication / selection / number of generations of domestication"

I have replaced with proportion of the domesticated strain. I do not think it can be 50% of the domestication/ selection/or generations of domestication, as this would be misleading. The hybrids have 50% of the genetic material from the domesticated strain, this is what I am trying to convey.

L304: again even though I understand by what you call "0-25-50-75-100%", I suggest using something else

Hopefully my explanations will have cleared this up a bit.

L362: could you add which studies you talk about here

Added.

Figure 2

I do not like your X-axis "percentage of domestication"; what does mean 100% of domestication or even 25-25-75... This has no sense for me even though I "understand" what you want to imply by that. I propose that you delete that and replace by number of generations in captivity or of selection, only indicate increasing domestication/selection. Also, I do not understand how the linear regression was performed? What values were used?

This is in contrast with reviewer 2 who likes this figure. Therefore, I have tried to clarify the sliding scale of domestication in the figure legend. Addition: "Percentage level of domestication used here is calibrated to our experimental design, with our domesticated strain being 100% domesticated, and our wild strain being 0% domesticated. Intermediate levels of domestication are produced through different combinations of hybridisation, be it direct hybrids (50% domesticated, 50% wild), wild backcrosses (25% domesticated, 75% wild) or domesticated backcrosses (75% domesticated, 25% wild). The linear regression was performed that was run between the percentage levels of domestication, including 0% (wild), 25% (wild BC), 50% (F1 and F2 hybrids), 75% (domesticated BC) and 100% (domesticated), and so should look like, this (below), but instead is overlaid onto the bar plots.

Table 1

This table is very important to understand the different strains studied. Yet it is hard to follow. Please modify it to make sure that we can see everything written and add a clear caption.

I think this is a problem with how the RSOS website compiles all of the files. Please find a better formatted table here:

Table 1 on page 34:

Origin			wet weight		heart weight		heart morphology (AHH, AHW, WHR)	
type	strain	location	freshwater	saltwater	freshwater	saltwater	freshwater	saltwater
domesticated	Mowi		♂ = 98 ♀ = 80	♂ = 72 ♀ = 93	♂ = 43 ♀ = 48	♂ = 34 ♀ = 42	♂ = 8 ♀ = 15	♂ = 12 ♀ = 15
hybrid	Figgjo (♀) × Mowi (♂)		♂ = 59 ♀ = 38	♂ = 52 ♀ = 32	♂ = 34 ♀ = 24	♂ = 24 ♀ = 16	♂ = 14 ♀ = 1	♂ = 13 ♀ = 9
	Mowi (♀) × Figgjo (♂)		♂ = 57 ♀ = 39	♂ = 53 ♀ = 43	♂ = 23 ♀ = 23	♂ = 16 ♀ = 16	♂ = 5 ♀ = 3	♂ = 6 ♀ = 8
F2 hybrid			♂ = 95 ♀ = 76	♂ = 92 ♀ = 98	♂ = 50 ♀ = 43	♂ = 45 ♀ = 43		
wild	Figgjo	58°81' N, 5°55' E	♂ = 103 ♀ = 81	♂ = 72 ♀ = 76	♂ = 54 ♀ = 52	♂ = 29 ♀ = 28	♂ = 13 ♀ = 12	♂ = 10 ♀ = 13
wild backcross			♂ = 90 ♀ = 78	♂ = 86 ♀ = 87	♂ = 36 ♀ = 41	♂ = 33 ♀ = 38		
domesticated backcross			♂ = 84 ♀ = 78	♂ = 92 ♀ = 79	♂ = 47 ♀ = 44	♂ = 45 ♀ = 30		

Supplementary table

This table is not visible in my pdf. Please modify and add a clear caption so it can be understood alone.

Again, I think this is a problem with how the RSOS website compiles all of the files. Please see below:

Supplementary table 1 Pairwise differences in mean wet body weight (kg) between the seven experimental strains, with significant ($p < 0.03$) contrasts between means highlighted in bold. Results are based on freshwater and saltwater body weight combined.

contrast	estimate	SE	df	t.ratio	p.value
Wild - Wild.BC	-0.1533	0.042991	29.14	-3.566	0.0193
Wild - Hybrid.FM	-0.31793	0.052619	29.07	-6.042	<.0001
Wild - Hybrid.MF	-0.38588	0.052478	28.76	-7.353	<.0001
Wild - F2	-0.3363	0.042933	28.99	-7.833	<.0001
Wild - Domesticated.BC	-0.43844	0.043013	29.2	-10.193	<.0001
Wild - Domesticated	-0.58184	0.043003	29.17	-13.53	<.0001
Wild.BC - Hybrid.FM	-0.16463	0.05259	29.01	-3.13	0.0539
Wild.BC - Hybrid.MF	-0.23258	0.052451	28.7	-4.434	0.0021
Wild.BC - F2	-0.183	0.042901	28.9	-4.266	0.0033
Wild.BC - Domesticated.BC	-0.28514	0.042982	29.11	-6.634	<.0001
Wild.BC - Domesticated	-0.42854	0.04297	29.08	-9.973	<.0001
Hybrid.FM - Hybrid.MF	-0.06795	0.060595	28.76	-1.121	0.9163
Hybrid.FM - F2	-0.01836	0.052546	28.91	-0.349	0.9998
Hybrid.FM - Domesticated.BC	-0.12051	0.052613	29.05	-2.29	0.2822
Hybrid.FM - Domesticated	-0.26391	0.052602	29.03	-5.017	0.0004
Hybrid.MF - F2	0.049582	0.052403	28.6	0.946	0.9613
Hybrid.MF - Domesticated.BC	-0.05256	0.052467	28.74	-1.002	0.9495
Hybrid.MF - Domesticated	-0.19596	0.05246	28.72	-3.736	0.0129
F2 - Domesticated.BC	-0.10215	0.042921	28.95	-2.38	0.243
F2 - Domesticated	-0.24555	0.042912	28.93	-5.722	0.0001
Domesticated.BC - Domesticated	-0.1434	0.042991	29.14	-3.336	0.0336

Supplementary table 2 Pairwise differences in mean wet body weight (kg) between sexes in the seven experimental strains, with significant ($p < 0.001$) contrasts between means highlighted in bold. Results are based on freshwater and saltwater body weight combined.

contrast	estimate	SE	df	t.ratio	p.value
female,Wild - male,Wild	0.080344	0.017711	2024.87	4.536	0.0005
female,Wild.BC - male,Wild.BC	0.084307	0.017542	2027.73	4.806	0.0001
female,Hybrid.FM - male,Hybrid.FM	0.010119	0.02436	2019.06	0.415	1
female,Hybrid.MF - male,Hybrid.MF	0.006145	0.023227	2019.06	0.265	1
female,F2 - male,F2	0.022058	0.016844	2019.8	1.31	0.9901
female,Domesticated.BC - male,Domesticated.BC	-0.00781	0.017617	2024.36	-0.443	1
female,Domesticated - male,Domesticated	-0.02227	0.017472	2022.86	-1.275	0.9923

Appendix C

Many thanks to both the reviewers and editor for their time, energy and comments, in what I know is a very testing time. Please find my responses below:

Minor comments

Line 33: might add “liver” here after “morphology”?

Added

Lines 47-49: please add the scientific name for “goats”, “tilapia” and “carp”

Added

Line 59: add “or liver” after “the heart”?

Added

Line 93: please check this sentence, should be slightly rephrase?

Rephrased

Line 113: add “also” before “fundamental”?

Added

Line 151: how many tanks were used for the saltwater experiment?

Two tanks – Added this to the text

Line 257: could you please explain and check how you obtain 2021? Should it be 1159 +1043 ?

I must apologise, those numbers for freshwater and saltwater were not updated early on. The numbers should read 1272 freshwater fish and 1146 saltwater fish, making a total of 2,418 fish. The reason that the denominator degrees of freedom in the model output looking at the effect of sex on body weight is different, is because not all of those 2,418 fish were able to be sexed (due to problems with identifying the genetic sex, either through failed PCRs or failed ‘sequencing’ etc.). 248 fish could not be genetically sexed, leaving 2,170 fish. You must then subtract the number of sample groups used in the model, and I believe this is where things become a bit more complicated. Because the degrees of freedom in an ANOVA are measured as the number of independent observations, and because there are random effects that confuse this, the lme4 and lmeTest package uses Satterthwaite approximation for degrees of freedom, and often these estimations of degrees of freedom are not perfect. When you include random effects, observations are taken to be correlated and not independent, and so the degrees of freedom depends on that correlation, and means that degrees of freedom cannot be calculated with ease. Some (a minority) suggest that p values and degrees of freedom should not be used for mixed effect models because of this. However, I feel that even if the estimate is slightly off, it is still more intuitive than having nothing there; especially when degrees of freedom are so universally understood. I hope this makes some sense?

Line 269: add a figure caption so it can be understood alone

Added

Lines 338-340: check the format of the number after F

Apologies! Fixed.

The Figure 3b and 3d are not cited in the text.

Added.

Table caption

Table 1: add a clear table caption (very hard to read this table in the MS); you can use the same information in the figure caption of figure 2.

Apologies for the bad formatting. I think the online portal does this when it converts my tables, that I upload as csv files, into a PDF.

Added: Table 1 Number of fish used for wet weight, heart weight and heart morphology measures such as adjusted heart height (AHH), adjusted heart width (AHW) and heart width-height residuals (WHR). Counts are broken down by life stage (freshwater or saltwater), experimental strains and sex (male = ♂ and female = ♀). **The seven experimental strains are included both in terms of their broader types, as well as the specific strains, the number of families that represent those strains, as well as the geographic location of the wild strain.**